# UNROLLING PALM FOR SPARSE SEMI-BLIND SOURCE SEPARATION

**Mohammad Fahes**[1]**, Christophe Kervazo**[1]**, Jérôme Bobin**[2] **& Florence Tupin**[1]
[1] LTCI, Télécom Paris, Institut Polytechnique de Paris, Palaiseau, France
[2] CEA Saclay, Gif-sur-Yvette, France

## ABSTRACT

Sparse Blind Source Separation (BSS) has become a well established tool for a wide range of applications – for instance, in astrophysics and remote sensing. Classical sparse BSS methods, such as the **P**roximal **A**lternating **L**inearized **M**inimization (PALM) algorithm, nevertheless often suffer from a difficult hyperparameter choice, which undermines their results. To bypass this pitfall, we propose in this work to build on the thriving field of algorithm unfolding/unrolling. Unrolling PALM enables to leverage the data-driven knowledge stemming from realistic simulations or ground-truth data by learning both PALM hyperparameters and variables. In contrast to most existing unrolled algorithms, which assume a fixed known dictionary during the training and testing phases, this article further emphasizes on the ability to deal with variable mixing matrices (a.k.a. dictionaries). The proposed Learned PALM (LPALM) algorithm thus enables to perform semi-blind source separation, which is key to increase the generalization of the learnt model in real-world applications. We illustrate the relevance of LPALM in astrophysical multispectral imaging: the algorithm not only needs up to $10^4 - 10^5$ times fewer iterations than PALM, but also improves the separation quality, while avoiding the cumbersome hyperparameter and initialization choice of PALM. We further show that LPALM outperforms other unrolled source separation methods in the semi-blind setting.

## 1 INTRODUCTION

**Blind source separation (BSS)** : In a wide range of scientific domains, including remote sensing (Dobigeon et al., 2013) and astrophysics (Picquenot et al., 2019), BSS is a very popular paradigm aiming at finding physical decompositions of multi-valued data (Comon & Jutten, 2010). Specifically, the data-set at hand is assumed to have been generated by a mixture of elementary signals, called sources. Assuming a linear mixing model, this mathematically writes as

$$X = A^*S^* + N. \tag{1}$$

The objective of BSS is then to recover the unknown mixing $A^* \in \mathbb{R}^{m \times n}$ and sources $S^* \in \mathbb{R}^{n \times t}$ matrices from the knowledge of the data set $X \in \mathbb{R}^{m \times t}$ only (and often despite the presence of a Gaussian noise $N \in \mathbb{R}^{m \times t}$). Unfortunately, it has long being emphasized that BSS is an ill-posed matrix factorization problem (Comon & Jutten, 2010): merely minimizing a data-fidelity term $\|X - AS\|_F^2$ over $A$ and $S$ (with $\|.\|_F$ standing for the Frobenius norm) often leads to spurious estimates which are very different from the true underlying $A^*$ and $S^*$ matrices. As such, most BSS approaches impose additional priors on the sought-after factors. Among them, sparsity has led to some of the state-of-the-art results (Zibulevsky & Pearlmutter, 2001; Bobin et al., 2007; 2015; Comon & Jutten, 2010). It amounts to tackling the following minimization problem:

$$\underset{A \in \mathbb{R}^{m \times n}, S \in \mathbb{R}^{n \times t}}{\arg\min} \quad \underbrace{\frac{1}{2}\|X - AS\|_F^2}_{\text{data-fidelity term}} + \underbrace{\lambda\|S\|_1}_{\text{sparsity constraint}} + \underbrace{\iota_{\{A^i, \|A^i\|_2^2 \le 1, i \in [1,n]\}}(A)}_{\text{oblique constraint}}, \tag{2}$$

where $\|S\|_1 = \sum_{i=1}^{n} \sum_{j=1}^{t} |S_{ij}|$, $\iota_B$ is the characteristic function of a set $B$, and $A^i$ denotes the i-th column of $A$. The first term is a data fidelity term stemming from the Gaussian-noise assumption,

the second one promotes sparsity while the last one – enforcing each column of $A$ to be in the $\ell_2$ unit ball – enables to discard degenerated solutions (in which $||A||_F \to \infty$ and $||S||_F \to 0$). In the following, we will specifically focus on sparse BSS.

Being a difficult multi-convex problem, the optimization strategy is key to minimize (2). Among the various iterative optimization algorithms that have been proposed so far, PALM (Attouch et al., 2010; Bolte et al., 2014) is particularly appealing due to its mathematical guarantees (Xu & Yin, 2013) because it has been proved to converge to a critical point of (2). An iteration of PALM for minimizing the cost function (2) consists in alternating the minimization with respect to $S$ and $A$. Specifically, at each iteration: i) a proximal gradient step of (2) is applied to update $S$ while $A$ is fixed; ii) $A$ is updated thanks to a proximal gradient step of (2) with $S$ fixed. The process is then repeated until the algorithm converges, see Section 2.

In the context of sparse BSS though, as highlighted in (Kervazo et al., 2020), PALM suffers from severe limitations, especially due to the high sensitivity of the estimated factors with respect to the choice of the regularization parameters. It is also emphasized in (Kervazo et al., 2020) that the solution strongly depends on the initialization. Altogether, these flaws, which are confirmed in our numerical experiments, undermine the results of PALM and limit its applicability in real-world applications. Therefore, we propose to investigate how algorithm unrolling methods can bypass these limitations. The general motivation is twofold : 1) introducing learned components within the original PALM algorithm provides a data-driven regularization of the estimated $A$ and $S$ factors. In particular, this can help to alleviate PALM difficult hyper-parameter choice. 2) Since (Kervazo et al., 2020) emphasizes the high impact of the optimization strategy on $A^*$ and $S^*$ estimate quality, learning some parameters of the gradient of (2) data-fidelity term is hoped to improve the results.

**Algorithm unrolling :** Algorithm unrolling (Monga et al., 2021) was pioneered in (Gregor & Le-Cun, 2010) for sparse coding. In our context, the sparse coding problem mostly corresponds to *non-blind* source separation, in which $A = A^*$ is known in (2). Quite similarly to PALM, such a cost function is usually minimized through an iterative proximal gradient step algorithm, such as the well known Iterative Shrinkage Thresholding Algorithm (ISTA) (Daubechies et al., 2004; Blumensath & Davies, 2008). Each ISTA iteration writes as:

$$\tilde{S}^{(k+1)} = \text{ST}_{\frac{\lambda}{L_S^{(k)}}} \left( \tilde{S}^{(k)} - \frac{1}{L_S^{(k)}} A^{*T}(A^* \tilde{S}^{(k)} - X) \right), \tag{3}$$

with $\tilde{S}^{(k)}$ the estimate of $S^*$ at iteration $k$, $L_S^{(k)}$ the Lipschitz constant of the gradient of (2) data-fidelity term and ST the soft-thresholding operator [1].

In (Gregor & LeCun, 2010), ISTA iterations are viewed as the layers of a recurrent neural network (RNN) with parameters $\theta = \frac{\lambda}{L}$, $W_1 = \mathbb{I} - \frac{1}{L} A^{*T} A^*$ and $W_2 = \frac{1}{L} A^{*T}$ (where $\mathbb{I}$ denotes the identity matrix). Instead of being fixed, the parameters $\theta, W_1, W_2$ are however trained, using backpropagation, from a data set. To make the distinction clearer in the following, we will write trainable variables with underlined letters, for instance $\underline{\theta}, \underline{W_1}, \underline{W_2}$. Assuming parameters to be untied across the layers, the $k$th layer of LISTA is thus given by:

$$\tilde{S}^{(k+1)} = \text{ST}_{\underline{\theta}^{(k)}} \left( \underline{W_1}^{(k)} \tilde{S}^{(k)} + \underline{W_2}^{(k)} X \right), \quad k = 0, 1, ..., K-1. \tag{4}$$

Consequently, the resulting network, named Learned ISTA (LISTA), "unrolls" ISTA and truncates it to $K$ iterations. The appeal of unrolled algorithms is twofold. On the one hand, they yield better estimates than iterative algorithms thanks to their higher representation power (Monga et al., 2021). On the other hand, numerous empirical results (Sprechmann et al., 2015; Wang et al., 2016; Zhang & Ghanem, 2018; Zhou et al., 2018) show that they generalize well to unseen samples, thanks to their interpretable structure.

Nevertheless, when the input distribution changes too much between the training and testing phases (e.g., $A^*$ changes, which is in particular the case in BSS), they often lead to impaired results as they tend to (over-)fit the training set distribution. To make the network more robust with respect to small changes in the dictionary $A^*$, it has been proposed to introduce small perturbations $\epsilon_A \sim \mathcal{N}(0, \sigma_{\max})$ on the elements of the dictionary $A^*$ during the training (Liu & Chen, 2019). Although such an attempt might be beneficial in some sparse coding problems (Zhao et al., 2011), the dictionaries vary much more in BSS because of the blind nature of the problem (and the variations might not be Gaussian). In contrast, we propose in this work to use a tailored PALM-based unrolling structure to account for the fact that $A^*$ is unknown and might largely change in the training and testing phases.

---

[1] $\text{ST}_\theta(v) = \text{sign}(v)\max(0, |v| - \theta)$, which applies component-wise.

## 1.1 RELATED WORKS

In source separation, algorithm unrolling has first been investigated for non-blind problems (Cowen et al., 2019) in which $A^*$ is known beforehand. For instance, (Qian et al., 2019) mostly unrolled ISTA into the same architecture as (Gregor & LeCun, 2010) in order to perform non-blind Hyperspectral Unmixing (HSU). In the same context, (Zhou & Rodrigues, 2021) unrolled the ADMM algorithm (Bioucas-Dias & Figueiredo, 2010). Recently, in an attempt to perform *blind* HSU, (Qian et al., 2020) also unrolled ISTA to estimate $S^*$, but proposed to add an extra layer in the neural network to learn $A^*$. The drawback is thus that the estimate $\tilde{A}$ is hard-coded in the network architecture, which is not well suited to the application on new hyperspectral images. Upon preparing this work, we became aware of the very recent work of (Xiong et al., 2021), which proposes an unrolled architecture in order to account for the spectral variabilities in $A^*$. They use an unsupervised cost function, which is a sum over the $K$ layers of the network of the data set reconstruction error ($X_{\texttt{train}}$ being the training set of size $t_{\texttt{train}}$, *i.e.* here a patch of the hyperspectral image):

$$\text{Loss} = \frac{1}{2t_{\texttt{train}}} \sum_{k=1}^{K} \|X_{\texttt{train}} - \tilde{A}^{(k)} \tilde{S}^{(k)}\|_F^2. \tag{5}$$

Such a loss is appealing as it does not require to know the ground truth factors $A^*$ and $S^*$ at training. A huge drawback is that the reconstruction-based error has long been advocated to possibly lead to spurious estimates, which are very different from the ground truth $A^*$ and $S^*$ factors (Comon & Jutten, 2010; Zibulevsky & Pearlmutter, 2001; Gillis, 2020). Consequently, (Nasser et al., 2021) rather added further priors on the sought after-factors in their unrolled algorithm cost function, at the price of adding extra hyperparameters to be tuned by hand. Since they focus on unrolling the multiplicative update (MU) algorithm (see also the earlier work of (Hershey et al., 2014) in this context), their network structure is quite different from both (Xiong et al., 2021) and the algorithm we investigate in this work.

Apart from the above articles, we would like to highlight that several works about unrolling methods for related-to-BSS problems, such as dictionary learning (Tolooshams et al., 2020) or even Magnetic Resonance Imaging (Arvinte et al., 2021), exist. However, as such problems have different purposes and challenges, it is beyond the scope of this article to review them.

## 1.2 MOTIVATIONS AND CONTRIBUTIONS

As stated above, current source separation state-of-the-art lets the practitioner in an unsatisfactory situation: 1) classical *Blind* source separation methods based on algorithms such as PALM are hindered by a cumbersome hyperparameter choice. Until now, unsupervised unrolling-based methods have felt short of solving such an issue: they often either lead to spurious solutions (as our numerical experiments will highlight) or still require parameters to tune by hand. 2) *Non-blind* source separation (NBSS) intrinsically assumes $A^*$ to be known and fixed during both train and test phases, which is often unrealistic in practical experiments.

In this article, we propose to merge the best of both worlds by performing *semi-blind* source separation (SBSS). Specifically, practitioners usually have access to an *imperfect* knowledge of the mixing matrix $A^*$, for instance by physics-derived simulators they might possess (which is in particular the case in astrophysics) or by approximate ground-truths of signals/images of the same nature as the ones they expect to observe (for instance, databases of spectra in remote sensing). The proposed Learned-PALM (LPALM) algorithm gives a principled and easy way to leverage such an imperfect knowledge to bypass the flaws of conventional sparse BSS algorithms. Specifically, the contributions of this article are:

- We unroll the PALM algorithm into the LPALM network, which is specifically tailored for sparse SBSS and to cope with *imperfectly known mixing matrices*. In particular, in contrast to usual unrolled algorithms such as LISTA (Gregor & LeCun, 2010) and LISTA-CP (Chen et al., 2018), which do not alternate between $A$ and $S$ updates, LPALM better accounts for the variability of the mixing matrix $A^*$ which occurs in source separation problems.

- We experimentally highlight the limitations of current unsupervised unrolled BSS algorithms: by leveraging the information contained within astrophysics simulations, LPALM largely outperforms the networks of (Xiong et al., 2021; Qian et al., 2020; Nasser et al., 2021) in terms of i) separation quality and ii) computation time. Compared to (Nasser

et al., 2021), the approach is different as it does not require the mixtures to be nonnegative, in contrast to the MU updates they use. This in particular enables to easily leverage sparsifying transforms such as wavelets.

- We provide an in-depth *comparative study between LPALM and PALM*. Specifically, we show that LPALM i) enables to automatically infer from the training set good regularization hyper-parameters and initialization; ii) accelerates the estimation time up to $10^4 - 10^5$ times. Altogether, these are key properties to make source separation reliable in real applications.

## 2 UNROLLING FOR SEMI-BLIND SOURCE SEPARATION

As evoked above, the PALM algorithm (Bolte et al., 2014) is a classical optimization algorithm in sparse BSS which sequentially and alternatively updates the factors $S$ and $A$ using proximal gradient steps. More precisely, for problem (2), PALM boils down to the following steps:

$$
\tilde{S}^{(k+1)} = \mathbf{ST}_{\theta^{(k)}} \left( \tilde{S}^{(k)} - \frac{1}{L_S^{(k)}} \tilde{A}^{(k)T}(\tilde{A}^{(k)}\tilde{S}^{(k)} - X) \right)
$$

$$
\tilde{A}^{(k+1)} = \Pi_{||.||_2 \leq 1} \left( \tilde{A}^{(k)} - \frac{1}{L_A^{(k)}}(\tilde{A}^{(k)}\tilde{S}^{(k+1)} - X)\tilde{S}^{(k+1)T} \right)
$$

(6)

where $\theta^{(k)} = \lambda/{L_S}^{(k)}$ and $L_S^{(k)}$ (*resp.* $L_A^{(k)}$) is usually chosen as the squared spectral norm of $A^{(k)}$ (*resp.* $S^{(k)}$). The proximal operator of the oblique constraint is denoted as $\Pi_{||.||_2 \leq 1}$. It corresponds to the projection on the $\ell_2$ unit ball of the columns of the considered matrix. A natural architecture to unroll PALM consists in keeping a similar alternating design, but different update types can be considered for $S$ and $A$. In the next subsection, we thus discuss the unrolling choices we did, the final LPALM algorithm being summarized in Subsection 2.2.

### 2.1 DERIVATION OF LPALM UPDATES

**Update step for $S$:** In the framework of PALM, the $\tilde{S}$ update is equivalent to a special case of an ISTA, which makes LISTA (4) the most straightforward trainable model to use. A key feature of LISTA is that it leverages a reparameterization of a standard proximal gradient descent update rule through the trainable parameters $\underline{W_1^{(k)}}$, $\underline{W_2^{(k)}}$ and $\underline{\theta^{(k)}}$. The advantage of using a LISTA-like update for $\tilde{S}$ would be twofold: i) learning $\underline{\theta^{(k)}}$ would enable to bypass PALM cumbersome hyperparameter choice ; ii) learning $\underline{W_1^{(k)}}$ and $\underline{W_2^{(k)}}$ would allow for extra degrees of freedom to efficiently estimate $S^*$. Nevertheless, in the LISTA framework, $A^*$ is assumed to be fixed across the training and test sets, which significantly deviates from the present SBSS case. This might lead to deteriorated estimates of $S^*$, since there is in particular every reason to believe $W_1^{(k)}$ and $W_2^{(k)}$ to be $A^*$-dependent (recall that initially LISTA reparametrization was motivated by $\overline{W_1 = \mathbb{I} - \frac{1}{L}A^{*T}A^*}$ and $W_2 = \frac{1}{L}A^{*T}$).

On the other hand, the empirical success of LISTA has motivated several works on its theoretical understanding (Moreau & Bruna, 2016; Giryes et al., 2018; Liu & Chen, 2019; Ablin et al., 2019). Of particular interest to us, (Chen et al., 2018) proved an asymptotic coupling of LISTA weights. Based on this, they derived a new architecture named LISTA-CP. In contrast to LISTA, using (4) as an update, a layer of LISTA-CP is given by:

$$
\tilde{S}^{(k+1)} = \mathbf{ST}_{\theta^{(k)}}(\tilde{S}^{(k)} + \underline{W^{(k)}}^T(X - A^*\tilde{S}^{(k)})), \quad k = 0, 1, ..., K-1.
$$

(7)

LISTA-CP explicitly makes use of $A^*$ and might thus in principle be better suited to deal with variable mixing matrices, provided that $A^*$ is known or at least decently estimated.

In Appendix D.1, we make a thourough experimental comparison between ISTA, LISTA, LISTA-CP, and ISTA-LLT (defined in the same section) in the presence of mixing matrix variabilities in the training and testing stages. These experiments confirm the insights described above : in a nutshell, LISTA does not accelerate the estimation of $S^*$ compared to ISTA for the same number of layers/iterations, and is also outperformed by LISTA-CP to a quite large extent, even when inaccurate $A^*$ are used in the updates. Consequently, our final choice is to opt for a LISTA-CP structure for $\tilde{S}$ update in the LPALM algorithm.

**Update step for $A$:**   The update of $A$ merely aims at minimizing a least-squares problem, with the constraint that each column of $A$ must belong to the $\ell_2$ ball of radius 1. As such, we merely propose to use an ISTA-LL (ISTA with Learned $L_A$) update where only the step size is trained. This is also motivated by the fact that a LISTA-CP-like architecture for $A$-update would make the network specific to an image/signal size, which is undesirable in real world applications.

**Applying LPALM in the testing phase – wavelet pre-processing:**   In addition to the two above steps, which are the core of LPALM, LPALM can be applied on pre-processed data $X$. For instance, in our realistic experiment of Subsection 3.2, we transformed $X$ into the isotropic undecimated wavelet domain (Starck et al., 2010). Specifically, the data $X$ is decomposed into the so-called coarse scale ${}^c X$ (*i.e.* large-scale approximation) and detail scales ${}^d X$ (*i.e.* fine-scale approximation). LPALM is then applied on ${}^d X$, which is sparser than ${}^c X$. As such a pre-processing is customary in sparse signal processing, and can be applied without loss of generality if the chosen wavelet domain is orthogonal, it is omitted in the algorithm summary of the next subsection; see Appendix A for more details.

## 2.2   SUMMARY OF THE LPALM ALGORITHM

**The LPALM algorithm:**   Putting together the above updates, the LPALM algorithm then reads as[2]:

$$
\boxed{
\begin{aligned}
&\text{Input: } X, \text{ output: } \tilde{A}_{\text{pred}} = \tilde{A}^{(K)}, \tilde{S}_{\text{pred}} = \tilde{S}^{(K)} \\
&\text{Initialize: } \tilde{A}^{(0)} = \left(1/\sqrt{m}\right)_{m \times n}, \tilde{S}^{(0)} = (0)_{n \times t} \\
&\text{for } k \text{ in } 0, ..., K-1 : \\
&\quad \tilde{S}^{(k+1)} = \text{ST}_{\underline{\theta^{(k)}}} \left( \tilde{S}^{(k)} - \underline{W^{(k)}}^T (\tilde{A}^{(k)} \tilde{S}^{(k)} - X) \right) \\
&\quad \tilde{A}^{(k+1)} = \Pi_{||.||_2 \leq 1} \left( \tilde{A}^{(k)} - \frac{1}{\underline{L_A}^{(k)}} (\tilde{A}^{(k)} \tilde{S}^{(k+1)} - X) \tilde{S}^{(k+1)T} \right)
\end{aligned}
}
\tag{8}
$$

where $\{\underline{\theta^{(k)}}, \underline{W^{(k)}}, \underline{L_A^{(k)}}\}_{k=\{0,...,K-1\}}$ are the $K(m \times n + 2)$ trainable parameters and the notation $(0)_{n \times t}$ denotes a matrix of size $n \times t$ filled with zeros.

On top of Subsection 2.1 discussion on $S$ and $A$ individual updates, using a LISTA-CP-like (*resp.* ISTA-LL-like) update for $S$ (*resp.* $A$) enables to keep the alternating structure of PALM. On the contrary, when using LISTA, $A^*$ (resp. $S^*$) appears only implicitly through the $\underline{W_1}^{(k)}$ and $\underline{W_2}^{(k)}$ matrices, which prohibits a direct alternation. To further justify the update choices we did, we numerically compare in Appendix D.2 the performances of the LPALM algorithm with other update steps (ISTA-LLT, LISTA-CP...).

**LPALM training loss function:**   In contrast to BSS unrolling methods (Xiong et al., 2021; Nasser et al., 2021), the semi-blind context allows to benefit from the knowledge of $S^*$ and $A^*$ *during the training phase*. It is then possible to define a loss which is sensitive to the quality of the predicted $\tilde{S}_{\text{pred}}$ and $\tilde{A}_{\text{pred}}$, such as the averaged sum of NMSE of both outputs:

$$
\text{Loss}_{\text{LPALM}} = \frac{1}{N_{\text{train}}} \sum_{i=1}^{N_{\text{train}}} \left( \frac{\||_i \tilde{S}_{\text{pred}} - {}_i S^*\||_F^2}{\||_i S^*\||_F^2} + \frac{\||_i \tilde{A}_{\text{pred}} - {}_i A^*\||_F^2}{\||_i A^*\||_F^2} \right),
\tag{9}
$$

where the lower-left subscript $i$ denotes the $i$-th sample of the considered (training) data set. It is important to point out that this loss does not require a fine tuning of regularization parameters as in the unsupervised algorithm of (Nasser et al., 2021). Additionally, since it is based on the predicted factors and not only the reconstruction error, the proposed LPALM will be shown in the experimental section to be more robust with respect to spurious solutions, which is essential to efficiently tackle non-convex problems. This is due to the fact that we leverage the knowledge contained within the training set, leading to some implicit regularization of the original problem. Implementation details are described in Appendix A. Among other, we would like to highlight that the training is easy: in particular, no fancy choice of the learning rate is needed (we used a constant value of $10^{-4}$ in all our

---

[2]https://github.com/mfahes/LPALM

experiments).

*Remark:* A major difference between our approach and the previous BSS unrolled algorithms (Qian et al., 2020; Xiong et al., 2021; Nasser et al., 2021) is the learning process itself. The latter are all designed to be trained and tested on a single matrix $X$: a part of its columns $\{X^i \in \mathbb{R}^{m \times 1}, i \in [1, N]\}$ is chosen to be the training set, and the model is then tested on the entire matrix $X$. This is due to the fact that problem (1) can be considered $t$ sub-problems of the form: $X^i = A^* S^{i^*} + N^i, i \in [1, t]$, with $X^i \in \mathbb{R}^{m \times 1}, A^* \in \mathbb{R}^{m \times n}, S^{i^*} \in \mathbb{R}^{n \times 1}$ and $N^i \in \mathbb{R}^{m \times 1}$. In contrast, LPALM takes a full matrix $X$ as a training or testing instance. The data sets are thus sets of matrices, not vectors.

## 3 NUMERICAL EXPERIMENTS

In this section, we assess the LPALM algorithm on astrophysical data simulations corresponding to X-ray images; see details in Appendix B. The data set used in Subsection 3.1 consists of 900 mixing matrices $A^* \in \mathbb{R}^{65 \times 4}$ and 900 matrices $S^* \in \mathbb{R}^{4 \times 500}$, which are split into 750 training samples and 150 testing samples. The mixtures $X = A^* S^* + N$ are corrupted with a white Gaussian noise $N$ so that SNR = 30 dB. The spectra (*i.e.* the columns of $A^*$) have large variabilities over the data set, see Figure 1. To mimic mildly sparse sources, the $S^* \in \mathbb{R}^{4 \times 500}$ matrices are simulated using generalized Gaussian distribution with a shape parameter $\beta = 0.3$ (which is usually a good proxy to mimic the distribution of wavelet coefficients of natural images). In Subsection 3.2, the same $A^*$ are used, but the sources in the test set come from real data (Picquenot et al., 2019); see Appendix B.

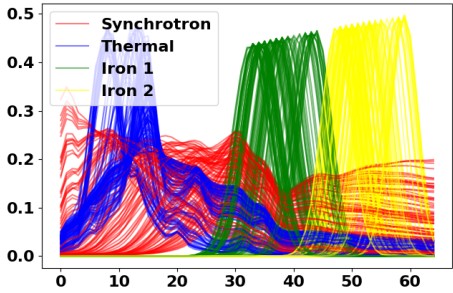

Figure 1: Illustration of the variability of the $A^* \in \mathbb{R}^{65 \times 4}$ matrices within the data set of Section 3. 100 examples of $A^*$ are depicted, each column being drawn with a different color.

In the following, the metric used to assess the separation quality will be the NMSE between the output of the network and the ground truth[3], computed either on the estimated sources $\tilde{S}_{\text{pred}}$ or the estimated mixing matrices $\tilde{A}_{\text{pred}}$:

$$\text{NMSE}(\tilde{S}_{\text{pred}}, S^*) = \frac{||\tilde{S}_{\text{pred}} - S^*||_F^2}{||S^*||_F^2} \text{ and } \text{NMSE}(\tilde{A}_{\text{pred}}, A^*) = \frac{||\tilde{A}_{\text{pred}} - A^*||_F^2}{||A^*||_F^2}. \qquad (10)$$

### 3.1 COMPARATIVE STUDY OF PALM AND LPALM

In this section, we specifically focus on comparing the PALM and LPALM algorithms to show that the latter enables to bypass PALM flaws. As emphasized in (Kervazo et al., 2020), the main pitfalls of PALM lie in its high sensitivity to the initialization and to the regularization parameter choice. On top of that, both are coupled, which makes PALM very hard to use in real-world applications. The NMSE in Subsections 3.1.1 and 3.1.2 is evaluated on the estimations $\tilde{S}$ of $S^*$.

### 3.1.1 LPALM OVERCOMES THE INITIALIZATION PROBLEM

In this subsection, we highlight that LPALM enables to bypass PALM sensitivity to the initialization. To do that, we launched PALM with 15 different initializations for a test sample: each time, a random

---

[3]When required, the usual permutation and scaling indeterminacy is corrected, as commonly done in BSS.

Table 1: Median NMSE of PALM over the test set, with the best $\lambda$ chosen from the training set, compared to that of LPALM.

| Algorithm | PALM | LPALM |
|---|---|---|
| **Median NMSE** | 0.0176 | 0.00085 |

matrix from the training set was chosen as an initialization $\tilde{A}^{(0)}$ of the algorithm. The sources were initialized as $\tilde{S}^{(0)} = A^{\dagger(0)}X$, with $A^{\dagger(0)}$ the pseudo-inverse of $A^{(0)}$. PALM was stopped when consecutive estimates verify both $||\tilde{S}^{(k+1)} - \tilde{S}^{(k)}||_F < 10^{-7}$ and $||\tilde{A}^{(k+1)} - \tilde{A}^{(k)}||_F < 10^{-7}$. On the other hand, LPALM was tested with the same initialization with which it was trained: $\tilde{A}^{(0)} = (1/\sqrt{m})_{m \times n}$, $\tilde{S}^{(0)} = (0)_{n \times t}$.

**Discussion**: With these settings, PALM median (over the initializations[4]) NMSE is 0.0201 (best: 0.0201, worse: 0.473). Furthermore, the standard deviation of PALM NMSEs with respect to the initialization, of 0.176, is huge compared to its median. This makes PALM highly unreliable in practice. On the contrary, LPALM obtained an NMSE of 0.0010 on the same sample. This means that, although launched from a very generic (and much worse than the one of PALM) initialization, LPALM managed to obtain better results than the best one of PALM.

### 3.1.2 LPALM OVERCOMES THE DIFFICULT HYPERPARAMETER CHOICE IN PALM

To try making PALM work in a realistic setting, we now infer both its initialization and regularization parameter from the training set. We then compare its results to those of LPALM.

As above, PALM is initialized with a mixing matrix $\tilde{A}^{(0)}$ taken at random from the training set, and $\tilde{S}^{(0)} = A^{\dagger(0)}X$, where $X$ is a data matrix from the test set to be factorized. However, the $\lambda$ hyperparameter is now chosen as the one minimizing the median NMSE between PALM source estimate $\tilde{S}_{\text{pred}}$ and $S^*$ over the training set. It is thus the best regularization parameter, among the 30 values we tried, that can be chosen from the training set.

**Discussion**: Figure 2(a) shows the results of PALM over the training set. Specifically, we see that the best regularization parameter is $\lambda = 2 \times 10^{-3}$, for which PALM reaches a median NMSE of $8.72 \times 10^{-3}$ in 1435 iterations. On the training set still, LPALM reaches better separation qualities than PALM, which is expected as it has more capacity to benefit from the learning. Table 1 displays the median results of both algorithms on the test set. Interestingly, LPALM generalizes well to unseen samples and maintains much better results than PALM, with a difference of more than one order of magnitude. As such, unrolling PALM enables to better leverage the knowledge of the training set than applying PALM with a good initialization and the best $\lambda$ value from the training set. Furthermore, we highlight that LPALM requires about 80 times fewer iterations than PALM (for the best $\lambda$ value, otherwise the ratio can rise even much more), making it much more tractable for large-scale data sets: on the test set, PALM required a median number of 2011 iterations, to be compared with the 25 layers of LPALM. For the sake of exhaustiveness, we furthermore plot in Figure 2(b) the evolution of the values of the NMSE across the layers of LPALM. It is interesting to note that in this experiment, LPALM seems to exhibit a quite monotonic improvement of its estimates over the layers, although this is not guaranteed and might not be the case for other data sets.

### 3.2 APPLICATION TO REALISTIC DATA

In this subsection, we compare LPALM to three unrolled BSS methods: MNNBU (Qian et al., 2020), SNMF-net (Xiong et al., 2021) and DNMF (Nasser et al., 2021). These unsupervised methods are all designed to be applied to a single $X$ matrix: a part of the columns of $X$ is chosen to be the training set, and the model is tested on the whole $X$. Concerning LPALM, the training set is created in a similar way as in the above section; see Appendix B. The methods are compared on a data set made of the same testing mixing matrices as in the above section, but this time realistic astrophysical

---

[4]To try to separate the impact of the initialization and the regularization hyper-parameter, we chose for each initialization the best $\lambda$ value. Please note that this would not be possible in real-life experiment, in which no ground-truth is available.

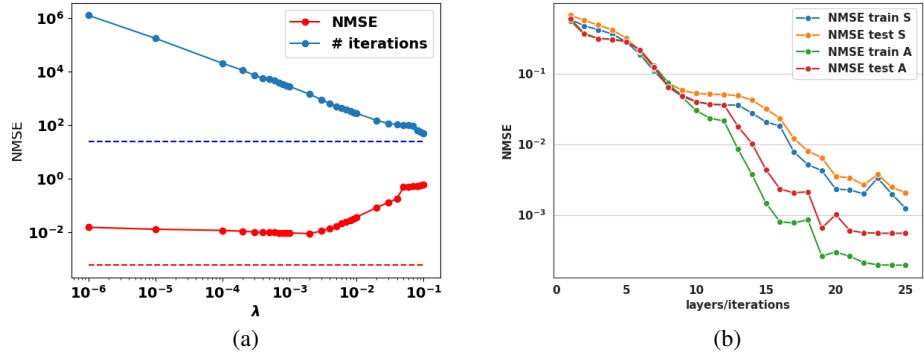

(a)                                    (b)

Figure 2: (a) PALM median NMSE and required number of iterations *w.r.t* $\lambda$ over $45$ samples of the train set; (b) LPALM average NMSEs per layer on both train and test sets.

sources are used; see Appendix B.

In addition, we had to slightly modify the data set to be able to perform fair comparisons with other unrolled architectures: indeed, MNNBU and SNMF-net use a sum-to-one constraint on $S^*$ columns. To use them in the proper way, we therefore normalized the $S^*$ matrix before applying these methods. We furthermore point out that these unrolled networks (as well as DNMF) assume the nonnegativity and sparsity of the sources in the *same* domain. The data are mildly sparse in the pixel domain. In LPALM, a wavelet-based pre-processing, which is customary for SBSS, is performed (see Appendix A for more details). Due to the non-negativity constraints for the other methods, it was impossible to transform the $X$ matrix in the wavelet domain when using them; they are applied in the pixel domain.

The results of LPALM applied to realistic data are shown in Figure 3. Qualitatively, all the four

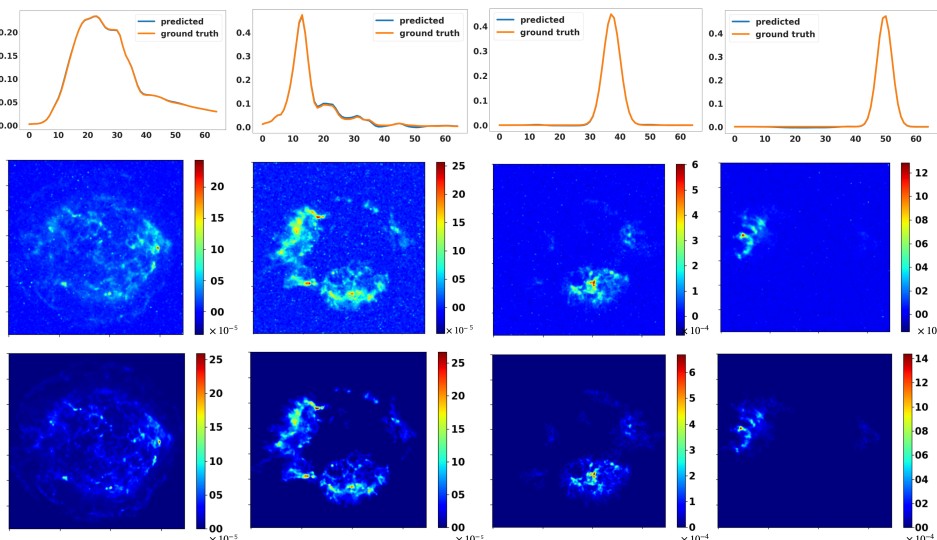

Figure 3: *First row:* LPALM estimate and ground truth of the four columns of the mixing matrix, *second row:* LPALM estimated sources, *third row:* ground truth sources.

estimated spectra are in very good agreement with the ground truth. Similarly, the sources are well estimated. The corresponding results of the other methods are shown in Appendix C (Figures 5, 6 and 7). Quantitatively, the comparison is performed in terms of the NMSE on the estimated mixing matrix $A$ because the ground truth sources are not the same for LPALM and other unrolled algorithms due to the column normalization. The results are shown in Table 2. The first line displays

Table 2: Comparison of LPALM with state-of-the-art methods in terms of NMSE over $A$

| Algorithm | MNNBU | SNMF-net | DNMF | LISTA | **LPALM** |
|---|---|---|---|---|---|
| NMSE($A_{\text{pred}}, A^*$) | 0.0962 | 0.2700 | 0.0466 | 0.0067 | **0.0002** |
| Generalization (model evaluation) | 0.4785 | 0.3053 | 0.5469 | 0.0088 | **0.0009** |
| Model retraining | 0.3181 | 0.3073 | 0.0922 | - | - |

the NMSE for the example of Figure 3, which clearly confirms LPALM superiority over other state-of-art methods. This is due to the fact that LPALM enables to leverage the knowledge contained within the whole training set (in particular, in the ground-truth), while the other methods only deal with single images. Furthermore, DNMF is better than MNNBU and SNMF-net: this is probably related to the fact that it uses a regularized cost function and not only the reconstruction error, which is less likely to lead to spurious solutions. We further benchmarked LPALM with the LISTA algorithm[5]. Although LISTA gives good results, it is largely outperformed by LPALM, which is probably due to the alternating structure of the latter, enabling to better deal with large variations over the $A^*$ matrices. Said differently, LPALM performs much better in the semi-supervised setting than LISTA.

We further assessed the generalization capacity of the four source separation methods. The second line of the table shows the median NMSE when the same learned models are applied on the other 149 examples of the test set. These results show that the three tested state-of-the-art methods do not generalize well to unseen samples. This is expected, as they are here trained on a single matrix and thus cannot learn the $A^*$ variabilities over the different matrices. This therefore demonstrates that the proper way to deal with $A^*$ variabilities with these networks is to re-train them for each new matrix $X$: this might be impractical for real-life applications, due to the computational cost and the difficult hyperparameter setting of some of them (learning rate schedule in particular). Lastly, the third line of Table 2 shows, over 20 samples, the LPALM three competitors' results when they are retrained for each new input matrix $X$. Interestingly, even in this setting LPALM largely outperforms the other methods, by almost two orders of magnitude. Therefore, not only LPALM does not need a retraining for each new image (as it is not trained on a single image), but it furthermore gives much better results than the other methods needing a retraining.

## 4 CONCLUSION

In this article, we introduce LPALM, an unrolled version of the PALM algorithm tailored for solving sparse SBSS problems *in the presence of variabilities of both generative factors $A^*$ and $S^*$*. This is key to apply unrolled sparse SBSS algorithms to real-world, where the mixing matrix is generally unknown but well constrained by the underlying physical phenomena. More precisely, the chosen unrolling architecture is thoroughly justified by both methodological considerations and experimental comparisons with state-of-the-art methods in the field. In contrast to available unrolled BSS methods that work in a purely unsupervised way without knowledge of the ground truth factors at training, the proposed LPALM algorithm provides *significantly better solutions*. Moreover, we illustrate that LPALM yields very *good generalization performances*, enabling to bypass PALM difficult initialization and regularization hyperparameter choice. Altogether, this makes LPALM a good unrolling-based candidate to analyze real-world data. To the best of our knowledge, this is *the first work using unrolling techniques for astronomical data*, and we believe that it would open much more perspectives to the introduction of such techniques in astronomy. Further research includes the application of LPALM to other domains, such as remote sensing.

## ACKNOWLEDGMENT

We would like to thank Fabio Acero for having provided the Chandra simulations.

---

[5]A few remarks are in order: 1) it was impossible to benchmark LPALM with the other considered algorithms such as LISTA-CP, as these require inside of their updates the knowledge of $A^*$; 2) LISTA does not explicitly estimate the mixing matrix $A^*$ but rather provides an estimation $\tilde{S}$ of the sources $S^*$. We therefore resorted to an ISTA algorithm to obtain $\tilde{A}$ from $\tilde{S}$ and $X$.

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

## APPENDIX

## A    IMPLEMENTATION DETAILS

For the training, Adam optimizer (Kingma & Ba, 2014) is used with $\beta_1 = 0.9$ and $\beta_2 = 0.999$. For the implementation, Pytorch library (Paszke et al., 2019) is used.

The training is done on 100 epochs with a learning rate LR=0.0001. The batch size is 1. We note that no fancy training is needed (in contrast to some of the other unrolled architectures we compared LPALM to).

**Non-Blind Setting:** The number of layers is $K = 25$. The parameters are initialized as follows:

- LISTA and LISTA-CP: 1) An $A$ is selected from the training set, 2) $L_S$ is taken as the largest eigenvalue of $A^T A$ , 3) $\theta^{k(0)} = \frac{\lambda}{L_S}$, $\lambda = 10^{-5}$, 4) $W_1^{k(0)} = \mathbb{I} - \frac{1}{L} A^T A$ (only for LISTA), 5) $W_2^{k(0)} = \frac{1}{L} A^T$. $\tilde{S}^{(0)} = (0)_{2 \times 500}$.

- ISTA-LLT: 1) An $A$ is selected from the training set, 2) $L_S^{k(0)}$ is taken as the largest eigenvalue of $A^T A$, 3) $\theta^{k(0)} = \frac{\lambda}{L_S^{k(0)}}$, $\lambda = 10^{-5}$. $\tilde{S}^{(0)} = (0)_{2 \times 500}$.

**LPALM:** The number of layers is $K = 25$. The parameters are initialized as follows: 1) 2 matrices $A$ and $S$ are selected from the training set, 2) $L_A^{k(0)}$ is taken as the largest eigenvalue of $SS^T$, for all $k \in [0, K-1]$, 3) $\theta^{k(0)} = \frac{\lambda}{L_S}$, for all $k \in [0, K-1]$, where $\lambda = 10^{-5}$, and $L_S$ is taken as the largest eigenvalue of $A^T A$, 4) $W^{k(0)} = \frac{1}{L_S} A$, for all $k \in [0, K-1]$, 5) $\tilde{S}^{(0)} = (0)_{n \times t}$, 6) $\tilde{A}^{(0)} = \Pi_{||.||=1}((1)_{m \times n})$.

In addition to the above parameters, we applied in the test phase of Section 3.2 experiment a wavelet pre-processing of the data $X$. Let us denote by $\Phi$ the transform from the direct to wavelet domains, $\Phi^{-1}$ is its inverse, $^c X$ the so-called coarse scale (i.e. large-scale approximation) and $^d X$ the detail scales (i.e. fine-scale). Let us further denote as $A^\dagger$ the pseudo-inverse of $A$. Once the model is trained, it is tested on the mixture $X$ as follows:

---

Input: $X$, output: $\tilde{A}_{\text{pred}} = \tilde{A}^{(K)}, \tilde{S}_{\text{pred}} = \tilde{S}^{(K)}$

Initialize: $\tilde{A}^{(0)} = \left(1/\sqrt{m}\right)_{m \times n}, \tilde{S}^{(0)} = (0)_{n \times t}$

Pre-processing:    $\Phi(X) = \{^c X, {}^d X\}$

*# Application of the learnt LPALM model*

for $k$ in $0, ..., K-1$ :

$$^d \tilde{S}^{(k+1)} = \text{ST}_{\theta^{(k)}} \left( {}^d \tilde{S}^{(k)} - \underline{W^{(k)}}^T (\tilde{A}^{(k)} \, {}^d \tilde{S}^{(k)} - {}^d X) \right)$$

$$\tilde{A}^{(k+1)} = \Pi_{||.||_2 \leq 1} \left( \tilde{A}^{(k)} - \frac{1}{\underline{L_A}^{(k)}} (\tilde{A}^{(k)} \, {}^d \tilde{S}^{(k+1)} - {}^d X) \, {}^d \tilde{S}^{(k+1)T} \right)$$

*# Recover the source coarse scale*

$^c \tilde{S}^{(K)} = \tilde{A}^{(K)\dagger} \, {}^c X$

*# Transform the source back to the direct domain*

$\tilde{S}^{(K)} = \phi^{-1}(\{^c \tilde{S}^{(K)}, {}^d \tilde{S}^{(K)}\})$

---

## B    DETAILS ABOUT THE ASTROPHYSICAL DATA

In this article, the mixing matrices come from astrophysical simulations, described in (Picquenot et al., 2019), which have been derived from real astrophysical data: the Cassiopea A supernovae remnant as observed by the X-ray space telescope Chandra `chandra.harvard.edu`. All the sources $S^*$ used are normalized.

Synthetic data used in Section 3.1

The $A^*$ matrices are composed of 4 different spectra of size 65 each: i) a synchrotron emission spectrum, ii) a thermal emission spectrum that is composed of various emission lines, which variabilities are similar and iii) two line emission spectra that are related to a single atomic component (e.g. iron) but with different redshifts due to the Doppler effect. The columns of the mixing matrices and their variabilities are displayed in Figure 1.

Realistic data used in Section 3.2

The kind of sources we focused on are remnants of supernovae; unfortunately, only few samples with a good estimate of the "ground truth" sources are actually available. We only have access to a single hyperspectral image of such an astrophysical object, which we use for testing in Section 3.2. For the whole data set, the $A^*$ matrices are the same as in Section 3.1. Training and test set however differ concerning the sources:

- **Test set**: In contrast to Section 3.1, we here assess LPALM when $S^*$ corresponds to real images of a supernovae remnant. Specifically, in order to be able to evaluate the different algorithm estimates, the data set $X$ is generated from ground-truth $A^*$ and $S^*$ matrices as follows:
    - The $A^*$ matrices are the same as in section 3.1.
    - The ground-truth sources $S^*$ are obtained from a real data set by the Generalized Morphological Component Analysis algorithm (GMCA – Bobin et al. (2007)), which is currently state-of-art in sparse source separation.

  In addition, dealing with these data is rather challenging since these sources exhibit partial correlations (pixels where more than one source is active with a large amplitude), which largely hampers the performances of most BSS methods (Bobin et al., 2015). To limit the impact of multiple active sources in $X$, we apply a randomization on the pixels of each source. Properly tackling the partial correlation problem using unrolled architectures is left for future research, but we still present LPALM results when such a randomization is not performed in Appendix D.3.

- **Train set**: in the absence of a large amount of training data for the sources $S^*$, we generated synthetic sources. More specifically, we generated the fine scales of the sources according to a generalized Gaussian distribution (GGD) with parameters $m$, $\sigma$, and $\beta$ (resp. mean, standard deviation and shape parameter). These parameters are fitted to the $tp$ non-zero elements of each true reference source ($t = 3000$ is the total number of pixels and $p$ is the proportion of non-zero elements). Then a realization of $tp$ samples of a GGD with the estimated parameters is simulated, the resulting vector is concatenated with $(1 − p)t$ zero elements. Finally, a random shuffling is applied to the vector. The GGD parameters are estimated from the histograms of non-zero elements of each source. This allows to simulate random sources that have approximately the same statistics as the original astrophysical sources in the wavelet domain. Such a simple process already enables to obtain a training set that is decent enough to enable LPALM to have good results in the test phase.

For both training and test sets, the data generation of $X$ was completed by adding a white Gaussian noise $N$ such that SNR = 30 dB.

## C  Details on section 3.2 experiments

The three methods we compare LPALM to are trained on a single matrix $X$. The training set consists in all the columns of $X$, which follows the observation of (Xiong et al., 2021) and (Qian et al., 2020) that the more column vectors used, the better the separation quality. Several hyperparameters (learning rate, number of epochs...) were tested in order to have the best performance *w.r.t.* the corresponding loss function.

**MNNBU**: The VCA (Nascimento & Dias, 2005) and FCLS (Heinz et al., 2001) algorithms are respectively used to initialize the $A$ and $S$ matrices, following the corresponding article recommendations. The initialization is shown in Figure 4. We see that the columns of $A^*$ are well extracted by the algorithm except for the synchrotron emission.

A learning rate schedule is used: $\text{lr} = 0.002$ at the first 99 epochs, and it is divided by 1.2 each 50 epochs starting from epoch 100. The total number of epochs for training is 500 epochs.

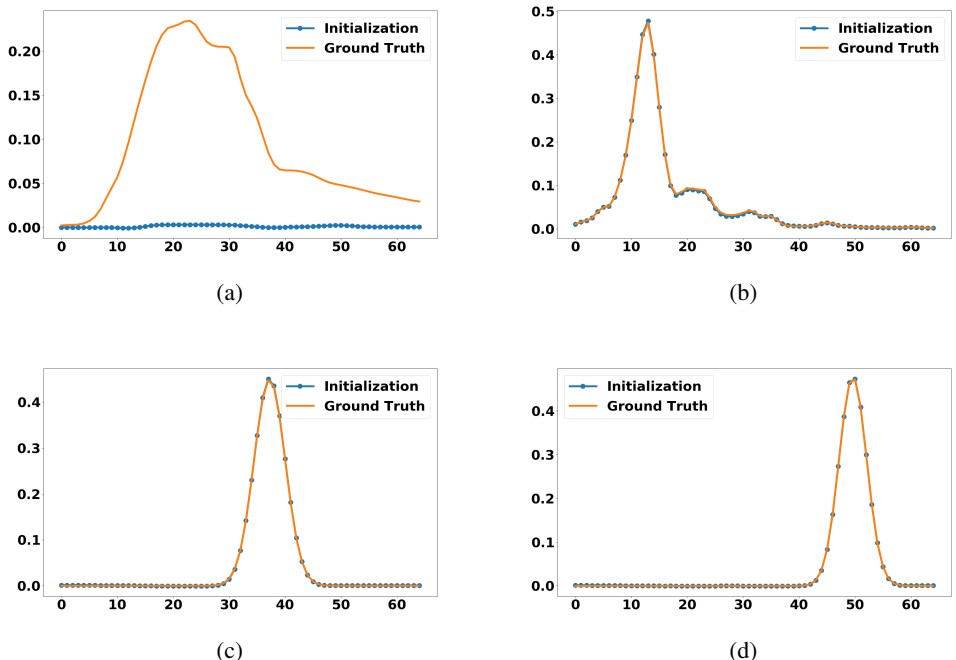

Figure 4: VCA results when applied to the data of Figure 3.

**SNMF-net**: The VCA (Nascimento & Dias, 2005) and FCLS (Heinz et al., 2001) algorithms are respectively used to initialize the $A$ and $S$ matrices, following the corresponding article recommendations; see Figure 4. The model hyperparameters are kept the same as the original paper code (learning rate schedules, number of layers...), except reducing the initial learning rate which was tuned in order to have the most possible decrease of the loss function.

**DNMF**: DNMF is initialized with matrices of ones (Nasser et al., 2021). The learning rate is $0.0008$, the number of epochs is $1500$, and we took the best regularization parameters among the ones tested in (Nasser et al., 2021). As such, DNMF might be slightly disadvantaged compared to MNNBU and SNMF-net, since these benefit from a much better initialization (note that however LPALM does not benefit from a better initialization than DNMF).

**Computation of the second line of Table 2**: The second line of Table 2 shows the losses of the models learned in the first line, when these models are evaluated on the whole testing set. The loss of MNNBU is trivially calculated using the ground truths $A^*$ and the predicted matrix $A_{\text{pred}}$, which is a hard-coded weight matrix in the network. For SNMF-net, the weights of the model are applied while initializing for each example with VCA and FCLS. For DNMF, the learned weights are used to predict $S$ from $X$, then a nonnegative least square algorithm is applied to predict $A$ from $X$ and DNMF output.

# D  ADDITIONAL EXPERIMENTS

## D.1  WHICH MODEL IS EMPIRICALLY THE BEST FOR UPDATING $S$ IN THE PRESENCE OF VARIABILITIES OVER $A^*$ ?

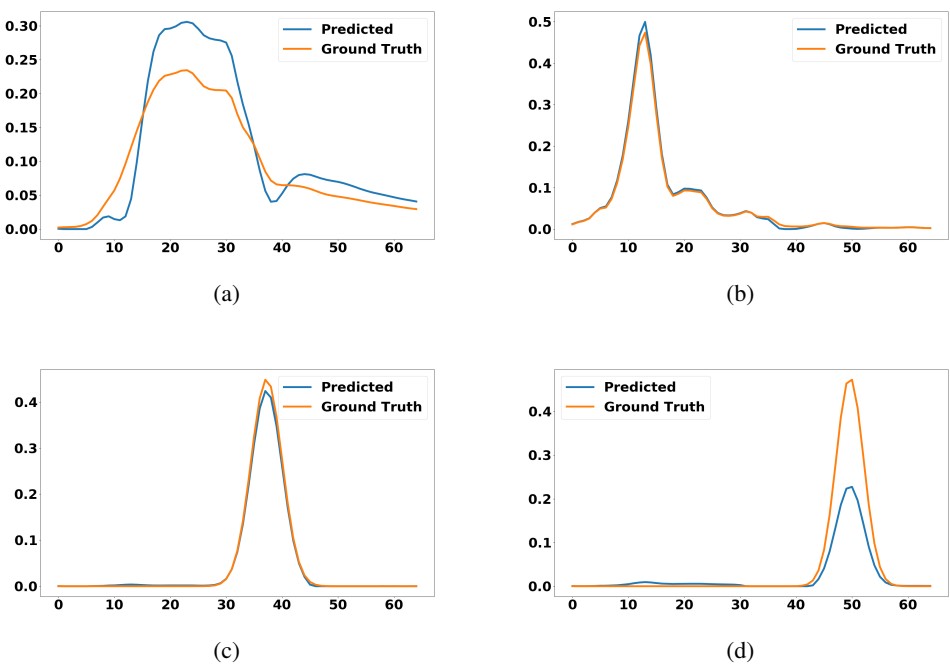

Figure 5: MNNBU predicted spectra from the data set of Section 3.2

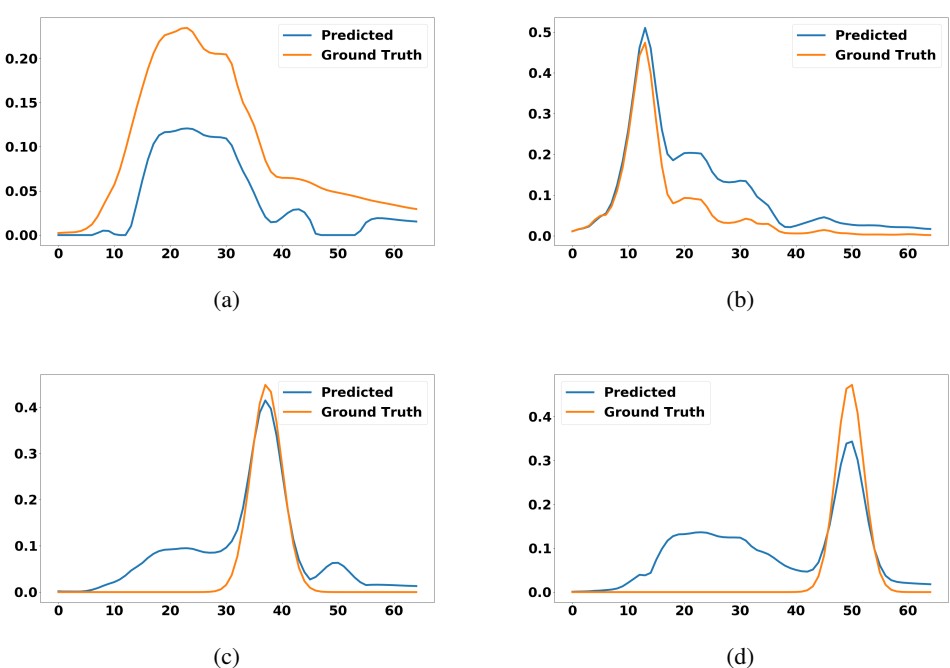

Figure 6: SNMF predicted spectra from the data set of Section 3.2

To experimentally confirm the unrolling choices we did for the $S$-update in Section 2.1, we assess the robustness of LISTA and LISTA-CP in the presence of mixing matrix variabilities at the training and testing stages. We also test ISTA-LLT (ISTA with Learned $L_S$ and Threshold), which is

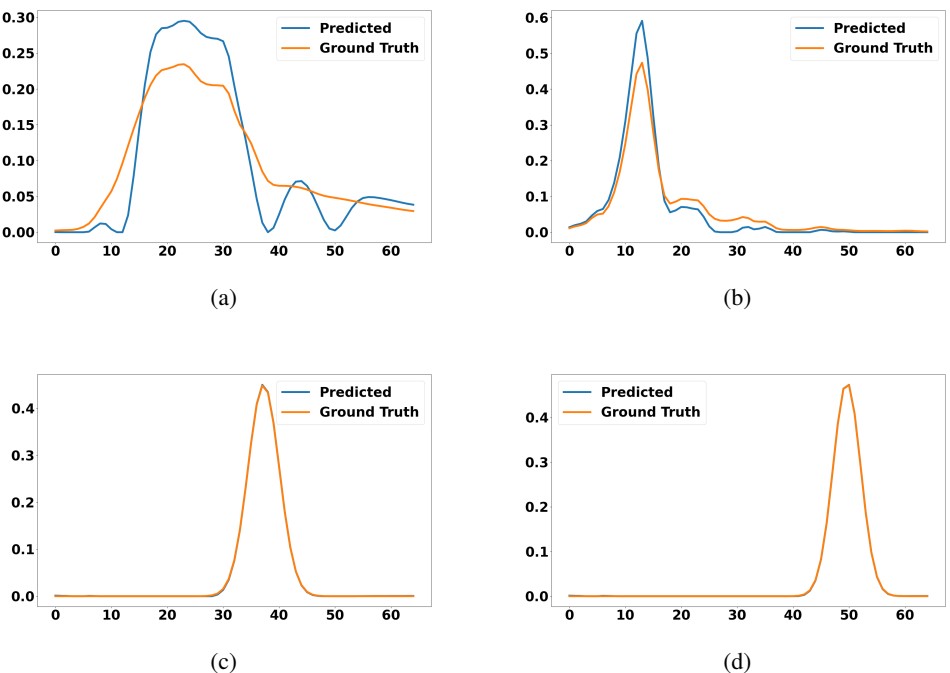

Figure 7: DNMF predicted spectra from the data set of Section 3.2

parameterized as follows:

$$\tilde{S}^{(k+1)} = \mathrm{ST}_{\underline{\theta^{(k)}}}\left( \tilde{S}^{(k)} - \frac{1}{\underline{L_S^{(k)}}} A^{*T}(A^* \tilde{S}^{(k)} - X) \right) \tag{11}$$

The goal of this subsection is twofold : 1) in the SBSS context, $A^*$ varies between the samples: we determine which of the tested unrolling methods is more robust in such a setting; 2) while the updates of LISTA-CP (Eq. 7) and ISTA-LLT (Eq. 11) require in principle the perfect knowledge of $A^*$ for each sample, in SBSS $A^*$ is unknown in the test phase and must be iteratively estimated within LPALM. Therefore, we evaluate LISTA-CP and ISTA-LLT when $A^*$ is perfectly estimated but also when it is not. The total number of layers for the unrolled methods is $K = 25$. The weights initialization is described in the Appendix A. To assess the quality of LISTA-CP and ISTA-LLT when $A^*$ is not perfectly known, a corrupted version of the mixing matrix $A^* + N_A$, with $N_A$ a white Gaussian noise such that the SNR with respect to $A^*$ is 10 dB, is used in LISTA-CP (7) and ISTA-LLT (11) updates. The corresponding algorithm results are denoted as LISTA-CP (SNR $= 10$ dB) and ISTA-LLT (SNR $= 10$ dB) in Figure 8.

ISTA truncated at 25 layers is used as a benchmark to compare the architectures. Its regularization parameter is $\lambda_{\mathrm{ISTA}} = 1.1 \times 10^{-5}$, which was chosen so as to minimize the average NMSE over the test set.

**Discussion**: Firstly, LISTA gives worse reconstruction qualities than ISTA truncated to 25 layers. This might be due to the fact that LISTA trainable parameters $W_1^{(k)}$ and $W_2^{(k)}$ are optimized over *all* the varying mixing matrices $A^*$ in the training set. Precisely, recall that LISTA reparametrization motivation was to set $W_1 = \frac{1}{L}A^{*T}A^*$ and $W_2 = \frac{1}{L}A^{*T}$. Being $A^*$-dependent, these parameters should rather be denoted as $W_1(A^*)$ and $W_2(A^*)$. When learnt from a single data sample $X$, it is thus likely that $W_1(A^*)$ and $W_2(A^*)$ are also $A^*$-dependent. When the training set is constituted of several samples in which the $A^*$ are not constant, the $W_1$ and $W_2$ learnt over the whole training set should resemble to an aggregate of the different $W_1(A^*)$ and $W_2(A^*)$. As such, in the test phase, LISTA parameters $W_1$ and $W_2$ might not be optimal for a given mixture $X$ which was generated from a specific new $A^*$. Such a phenomenon also probably explains the superiority of LISTA-CP and ISTA-LLT over LISTA, since, for a given $X$, they precisely use information about the mixing

through the presence of $A^*$ within their updates, which enhances their robustness with respect to the variabilities of $A^*$.

As expected, the results of LISTA-CP and ISTA-LLT both deteriorate when noisy mixing matrices are used within their updates (LISTA-CP (SNR = 10 dB) and ISTA-LLT (SNR = 10 dB) curves). While the results of ISTA-LLT (SNR = 10 dB) become worse than ISTA, LISTA-CP (SNR = 10 dB) still gives better estimates than ISTA. The higher ability of the LISTA-CP architecture to deal with noisy $A^*$ estimate might be linked to a higher capacity of the network than ISTA-LLT, as there is a much higher number of parameters. Such a property justifies the choice of a LISTA-CP like update in LPALM.

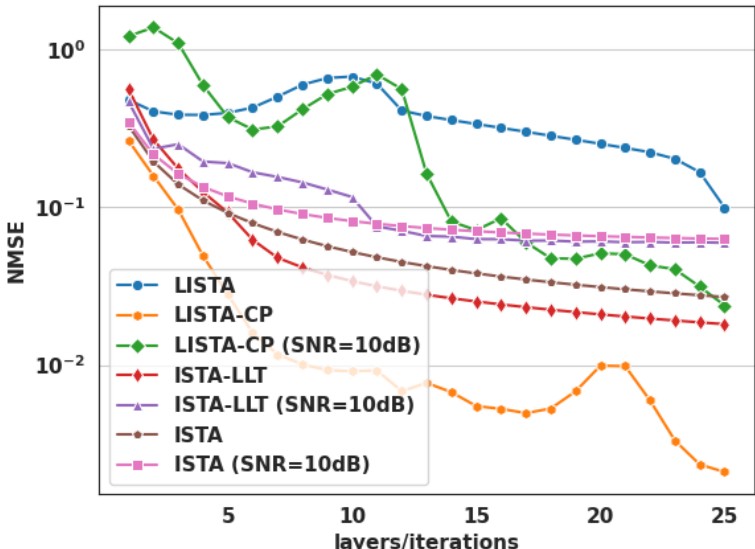

Figure 8: NMSE of several unrolled non-blind architectures, compared to ISTA truncated to 25 layers. The LISTA-CP and ISTA-LLT curves correspond to the updates (7) and (11) where $A^*$ is perfectly known. The LISTA-CP (SNR = 10 dB), ISTA-LLT (SNR = 10 dB) and ISTA (SNR = 10 dB) curves correspond to updates using a noisy estimate of $A^*$.

## D.2 TEST OF THE POSSIBLE UPDATES WITHIN LPALM

In D.1 we assessed various updates strategies for $S$. In this subsection, we try different update forms *within the alternating structure of LPALM*. In this work, we considered five update strategies: LISTA, LISTA-CP, ISTA-LLT, ISTA-LL or an explicit update (using the closed form gradient of (2) data-fidelity term). However, some update forms should be ruled out when unrolling PALM:

- Any use of LISTA update, either for $A$ or $S$, would prohibit an alternating structure of the algorithm;

- For $A$, ISTA-LLT does not have any sense, since there is no threshold to consider. LISTA-CP cannot be considered either, as it would imply a fixed number of pixels, which is impractical in real-life experiments;

- For $S$, both the explicit and the ISTA-LL updates require to tune the regularization parameter $\lambda$ by hand, which is impractical as attested in Kervazo et al. (2020).

Furthermore, an explicit update of both $A$ and $S$ corresponds to the PALM algorithm, which has already been tested in Section 3.1. The remaining possibilities are assessed in Table 3, on the same data set used in Section 3.1. It can be seen that LPALM (first column) largely outperforms its counterparts using ISTA-LLT updates for $S$. This is likely to be linked to the higher number of

Table 3: Comparison of the different possible updates within an unrolled PALM architecture. The median NMSE over the test set is displayed. The first column corresponds to the LPALM algorithm.

| S update | LISTA-CP | LISTA-CP | ISTA-LLT | ISTA-LLT |
| A-update | ISTA-LL | Explicit | ISTA-LL | Explicit |
|---|---|---|---|---|
| Median NMSE($S_{\text{pred}}, S^*$) | 0.00085 | 0.00097 | 0.7943 | 0.7948 |

parameters in the LISTA-CP updates, enabling LPALM to have a higher representation power.

It is interesting to note that an explicit update for $A^*$ leads to very similar results as LPALM in this experiment. We preferred to keep within LPALM a learned component for $A$ update, which might be beneficial in experiments where $A^*$ could be more difficult to learn (for instance, a larger matrix $A^*$ would imply a larger number of elements to estimate). Furthermore, as learning the step sizes inside of the $A$ update only encompasses $K$ parameters to learn, the additional training cost is often negligible (as well as the over-fitting risk).

### D.3 LPALM SEPARATION RESULTS IN PRESENCE OF PARTIAL CORRELATIONS

The original astrophysical data exhibit partial correlations, which are known to dramatically hamper the performances of most SBSS methods (Bobin et al., 2015). For that purpose, we previously presented results where the sources are randomized, to avoid such partial correlations. Figure 9 shows the results of LPALM when applied to the raw astrophysical data. These results are of reasonable quality but show leakages that directly originate from the presence of these partial correlations. Since the training set used to train LPALM does not contain such partial correlations, it cannot learn an unmixing procedure that is robust to this type of artifacts. An in-depth comparison with other unrolled algorithms in the presence of partial correlations, as well as properly tackling this issue within LPALM, are left for future work.

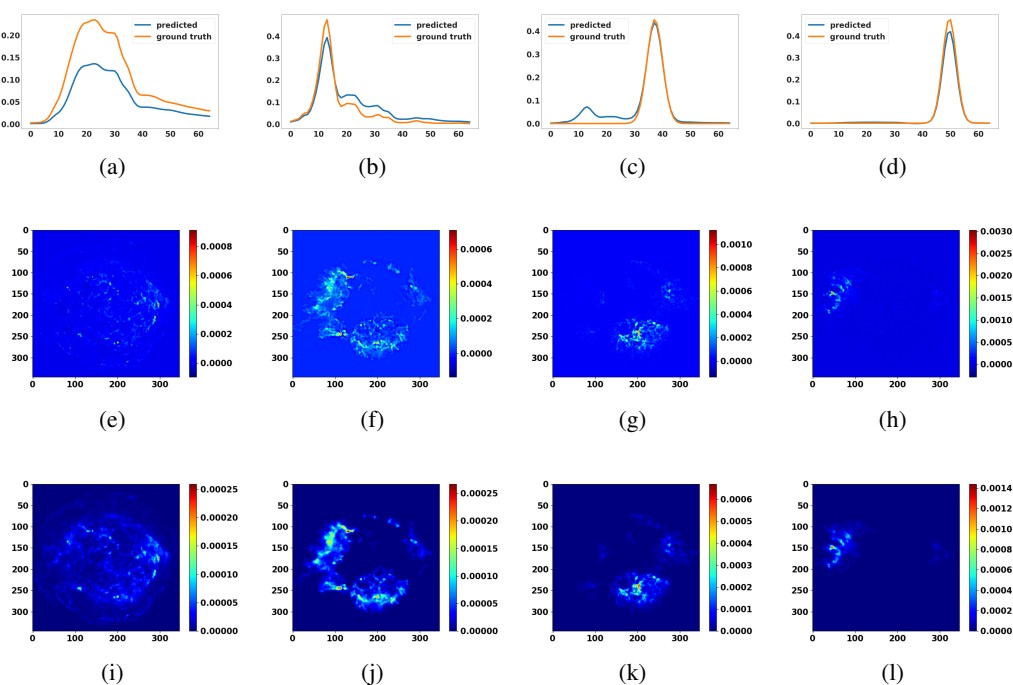

Figure 9: Prediction vs ground truth of: From (a) to (d) the mixing matrix, from (e) to (h) the sources (the ground truth sources are shown from (i) to (l))

### D.4 IMPACT OF THE NUMBER OF LAYERS IN LPALM

In this subsection, we study the impact of LPALM number of layers on the final estimate quality. The NMSE of LPALM output is plotted in figure 10 as a function of the number of layers $K$. As can be seen from the plot, LPALM first better estimates the sources and the mixing matrices when the number of layer increases, which is expected as the network representation power increases. The best $K$, namely $K = 50$, is very low compared to the usual number of PALM iterations; see Figure 2(a). Nevertheless, when $K$ becomes too large, the separation quality decreases due to over-fitting. In practice, choosing $K$ values leading to over-fitting is easily avoided since the practitioner can monitor the evolution of the validation loss, as is generally done.

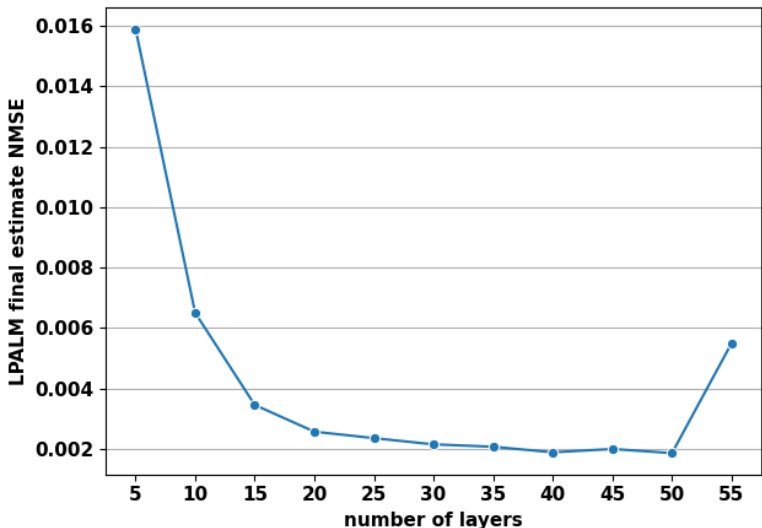

Figure 10: Evolution of the sum of the NMSE over the $A$ and $S$ estimates predicted by LPALM as a function of the network number of layers $K$.

### D.5 IMPACT OF THE DIFFERENT TERMS IN LPALM LOSS FUNCTION

For the sake of completeness, we here perform an ablation study on LPALM loss function. We test four different losses. The results are reported in Table 4.

- *Supervised on $S$ and $A$*, which corresponds to LPALM loss function (9):

$$\frac{1}{N_{\text{train}}} \sum_{i=1}^{N_{\text{train}}} \left( \frac{\|_i\tilde{S}_{\text{pred}} - {_i}S^*\|_F^2}{\|_i S^*\|_F^2} + \frac{\|_i\tilde{A}_{\text{pred}} - {_i}A^*\|_F^2}{\|_i A^*\|_F^2} \right)$$

- *Supervised on $S$ only*, which corresponds to the loss:

$$\frac{1}{N_{\text{train}}} \sum_{i=1}^{N_{\text{train}}} \left( \frac{\|_i\tilde{S}_{\text{pred}} - {_i}S^*\|_F^2}{\|_i S^*\|_F^2} \right)$$

- *Supervised on $A$ only*, which corresponds to the loss:

$$\frac{1}{N_{\text{train}}} \sum_{i=1}^{N_{\text{train}}} \left( \frac{\|_i\tilde{A}_{\text{pred}} - {_i}A^*\|_F^2}{\|_i A^*\|_F^2} \right)$$

- *Unsupervised*, which corresponds to using as loss a data-fidelity term:

$$\frac{1}{N_{\text{train}}} \sum_{i=1}^{N_{\text{train}}} \left( \frac{\|_i X - {_i}\tilde{A}_{\text{pred}} \, {_i}\tilde{S}_{\text{pred}}\|_F^2}{\|_i X\|_F^2} \right)$$

Table 4: Comparison of the estimation quality yielded by LPALM with four different losses.

| Loss | Unsupervised | Supervised on S | Supervised on A | Supervised on S+A |
|---|---|---|---|---|
| Median NMSE$(S_{\text{pred}}, S^*)$ +NMSE$(A_{\text{pred}}, A^*)$ | 0.568 | 0.00115 | 0.00191 | 0.00112 |

This experiment highlights that using a supervised cost function leads to much better results than merely using an (unsupervised) reconstruction error. Such a phenomenon, which could already be hinted by the results in Section 3.2 of MNN-BU and SNMF-net, is well known in blind source separation, in which the data-fidelity term is often complemented by additional regularizations (Comon & Jutten, 2010). Please note that there might be better ways to perform unsupervised learning. Such an extension of LPALM is however beyond the scope of this work.

Concerning the supervised cost functions we consider, all of them mostly obtain similar results. As such, we used in this work the sum of the NMSE over $A$ and $S$, which might be the most general one as it does not put any emphasis on either term. According to the considered experimental setting, the practitioner might nevertheless prefer another loss: for instance, it can be hinted that if one has access to a good simulator for either $A^*$ or $S^*$ in the training phase, a loss adding more emphasis on the corresponding term might lead to better results. More generally, investigating further the impact of the loss function is left for future work.

