# OpenReview forum: "Unrolling PALM for Sparse Semi-Blind Source Separation"
_ICLR.cc/2022/Conference — ICLR 2022 Poster_

### Official Review · Reviewer_fMrf · 2021-11-01

**Correctness:** 3
**Technical Novelty And Significance:** 2
**Empirical Novelty And Significance:** 2
**Recommendation:** 8
**Confidence:** 4

**Main Review:**

Strengths

- Unrolling implementation of the PALM algorithm building on existing works e.g. LISTA-CP and ISTA-LL. The authors aim at overcoming some weaknesses of previous approaches such as the need of tuning parameters, convergence  to spurious local minima, etc.
- This is the first work that  studies unrolled formulation of BSS problem on  hyperspectral astronomical data unmixing and show promising results of the proposed L-PALM algorithm.

Remarks/Weakness
- The authors claim that the loss function they use is more robust w.r.t. spurious solutions. However, there is no empirical evidence showing that this statement is true. For that to be clear, the authors should present an experiment that would compare their approach when a)  the loss function in (5)  vs b) the proposed loss function in (9) is used.
- In Section 3.1 the authors compare LISTA-CP approach that is adopted for updating S vs other unrolled ISTA versions. However, the experiment presented there is not quite convincing: First the number of the sources in set to 2 which is quite small and do not correspond to a realistic number in practical applications. Second, the authors present the value NMSE as it evolves for a maximum of 5 layers. This is not quite insightful to draw safe conclusions since virtually all algorithms keep decreasing at layer=5. The plot in Figure 2 should thus show NMSE values for a larger number of layers.
- In Section 3.2.1, the authors use what they a call as "bad" initialization for both PALM and L-PALM concluding that the latter is robust to bad initializations. However, the use of a single "bad" initialization could be potentially misleading. The authors could perform experiments for different random initializations and then measure median NMSE for providing more convincing arguments.
- The paper lacks any theoretical argument that would give an explanation of the superiority of the proposed approach relative to state-of-the-art.

Typo:
 -  page 3, 1st bullet on Contributions: tailored instead of taylored




**Summary Of The Paper:**

The authors proposed a new deep unrolling version of the proximal alternating minimization (PALM) algorithm for addressing the semi-supervised blind source separation problem.  The main contribution of the proposed approach over relevant state-of-the-art algorithms is the use of the sum of NMSE losses of the two matrix factors and the specific ISTA-type updates that are implemented at each layer of the network for updating each matrix factor. The authors provide empirical results that demonstrate the superior performance of the proposed approach over other unrolling based state-of-the-art BSS algorithms.

**Summary Of The Review:**

The paper provides an unrolling version of the well-known PALM algorithm with application to blind source separation for unmixing of hyperspectral astronomical data. The authors provide empirical results that show the advantages of the proposed approach over other relevant state-of-the-art unrolling algorithms. Given the existing state-of-the-art algorithms the technical  novelty of the current work is limited. Yet, the work presented could be of practical interest when it comes to the BSS application that is studied. However, the result are not always convincing (see Weaknesses) and could be significantly improved. The paper also lacks any theoretical justification and the evaluation of the method in only through the empirical results provided.  The paper could be also better organized since I feel the authors use a lot of subsections (especially in the experimental part), which disrupt the flow of the paper.

------------------------------------------
Post-rebuttal update

The authors have addressed my comments and concerns adding further experiments and re-organizing the paper. I, therefore, raise my score.

---

> ### Author Response · Authors · 2021-11-19
> **Response to Reviewer fMrf**
>
> We thank the reviewer for this detailed review. We will soon upload an updated version of our work.
>
> > - Unrolling implementation of the PALM algorithm building on existing works e.g. LISTA-CP and ISTA-LL. The authors aim at overcoming some weaknesses of previous approaches such as the need of tuning parameters, convergence to spurious local minima, etc.
> >- This is the first work that studies unrolled formulation of BSS problem on hyperspectral astronomical data unmixing and show promising results of the proposed L-PALM algorithm.
>
> Thank you for this positive feedback.
>
> >The authors claim that the loss function they use is more robust w.r.t. spurious solutions. However, there is no empirical evidence showing that this statement is true. For that to be clear, the authors should present an experiment that would compare their approach when a) the loss function in (5) vs b) the proposed loss function in (9) is used.
>
> This is a good remark. First, please note that MNN-BU, SNMF-net and DNMF are unsupervised algorithms, and that our algorithm outperforms them all. Nevertheless, this could also be due to the differences in the neural network itself. This is why, to answer your remark, we will add in the new version of the article a new NMSE curve with the unsupervised cost function (5). The results, that we already obtained, highlight that our supervised approach clearly performs better.
>
> >In Section 3.1 the authors compare LISTA-CP approach that is adopted for updating S vs other unrolled ISTA versions. However, the experiment presented there is not quite convincing: First the number of the sources in set to 2 which is quite small and do not correspond to a realistic number in practical applications. Second, the authors present the value NMSE as it evolves for a maximum of 5 layers. This is not quite insightful to draw safe conclusions since virtually all algorithms keep decreasing at layer=5. The plot in Figure 2 should thus show NMSE values for a larger number of layers.
>
> To address both points :
> - If time allows, we will raise the number of sources to 4 in the future version of our work, to increase the experiment difficulty.
> - We will also increase the number of layers. Please note however that it is part of the work to propose a faster architecture than ISTA / PALM, and as such it is already interesting to compare the algorithm results after few iterations / layers.
>
> >In Section 3.2.1, the authors use what they a call as "bad" initialization for both PALM and L-PALM concluding that the latter is robust to bad initializations. However, the use of a single "bad" initialization could be potentially misleading. The authors could perform experiments for different random initializations and then measure median NMSE for providing more convincing arguments.
>
> This is indeed an interesting point. Please remark the following:
> - Concerning LPALM, the algorithm has been trained with a generic initialization (the initial $A$ being a matrix filled with $1$ values and the initial $S$ with zero values). Changing such an initialization in the testing phase might lead to deteriorated results, since it differs from the training. As such, the strength of LPALM is not to enable the user to use various initialization, but rather to propose a generic initialization, which works well for new data $X$.
> - Concerning PALM, we could indeed try several random initialization, to show that the algorithm is on average outperformed by LPALM. If time allows, we will try to add such an experiment in the future version of our work. Nevertheless, we did not include it in our original submission because PALM has already been shown to be sensitive to the initialization in *[1] Kervazo, C., Bobin, J., Chenot, C., \& Sureau, F. (2020). Use of PALM for $\ell_1$ sparse matrix factorization: Difficulty and rationalization of a two-step approach. Digital Signal Processing, 97, 102611.*. Therefore, we were not sure about the novelty that would be explored by such an experiment.
>
> >The paper lacks any theoretical argument that would give an explanation of the superiority of the proposed approach relative to state-of-the-art.
>
> This is true, although we still aimed at proposing a quite in-depth understanding of our algorithm (this is in particular the purpose of Section 3.1, which aims at focusing on a subpart of LPALM algorithm to better understand its behavior).
>
> >page 3, 1st bullet on Contributions: tailored instead of taylored.
>
> Thanks for the remark.
>
> >The paper could be also better organized since I feel the authors use a lot of subsections (especially in the experimental part), which disrupt the flow of the paper.
>
> In the future version of our work, we will try to make the organization clearer (by removing the details and transferring them to the Appendices for instance).

---

### Official Review · Reviewer_bWys · 2021-11-02

**Correctness:** 4
**Technical Novelty And Significance:** 3
**Empirical Novelty And Significance:** 3
**Recommendation:** 8
**Confidence:** 4

**Main Review:**

Strengths:
1] I appreciate that you explore and validate the details of your LPALM parameterization (i.e. section 3.1), which seems to often get left out in unrolling works

Other Suggestions:

1] your related works section is good, but you may wish to consider the following papers which also address non-blind source separation with unrolling:
Cowen et al, "LSALSA: accelerated source separation via learned sparse coding."
Hershey et al, "Deep Unfolding: Model-based Inspiration of Novel Deep Architectures"

2] in section 1.2 you say that "[NBSS] is limited by the fact that most methods assume A* to be known and fixed...". Isn't that the definition of NBSS?

3] I don't think you broke any rules, but for example in Sec 3.1, the adjustment of line-to-line margins (and font sizes?) is too obvious that you are cramming information. I suggest trying to pare down your experimental description and maybe put some stuff in the appendix (for a specific example, instead of narrating all observations, just highlight the conclusions). Then in Section 3.3 you can give readers the really important information about preprocessing.
a) I want to emphasize that this isn't just aesthetic, but when things are crammed it's harder for readers to gather the important details. I am having a hard time understanding how your competitor methods are trained on "a part of the pixels of the image". This seems like a completely different training paradigm that confounds your comparison.

4] In Fig 3b, instead of showing the intermediate results I think it is more interesting to show different sized LPALMs (how much NMSE gain do you get from training a longer recurrent net?). I also suggest using different markers for each competitor so that people don't have to rely on similar colors.

**Summary Of The Paper:**

Problem: in the context of blind source separation (BSS), the PALM algorithm is highly sensitive to regularization parameters and initialization.
Solution: use algorithm unrolling to learn the sensitive (hyper-)parameters.
Results: LPALM outperforms PALM, and is compared to some competitors*.


**Summary Of The Review:**


1) The reason I gave a 2/4 in "Correctness" is that, while you do solve the sensitivity problem w.r.t. PALM's hyperparmeters, you introduce a slew of new hyperparameters associated with training. So the real question is: How does the sensitivity to those new hyperparameters compare? (I'm not saying you're incorrect, but saying it's not well-supported that the sensitivity problem is solved; you even mention that learning rate schedule was a "difficult" hyperparameter to set in Sec 3.3)
2) In terms of technical novelty or significance, I think this is standard unrolling. The thing I have not seen before is incorporating priors for both A* and S* both into the loss function.
3)  To be clear, unrolling is fully understood to give improvements over a non-learned counterpart (although it is good to rehash it in specificity in Sec 3.1,3.2, they are not sufficient for publication in ML). So when it comes to Section 3.2, the main problem I have is that it seems the other methods were trained very differently. My main takeaway is that the overall procedure gives an improvement, which is good (or even great), but in terms of the significance I have to defer to someone who knows about this application.

Edit: I want to thank the authors for giving thorough feedback to each comment and adding/updating the appendix comprehensively. I am satisfied with the responses and have raised my score accordingly. I see now that the learning-rate-schedule difficulty is a comment on the other methods.

---

> ### Author Response · Authors · 2021-11-19
> **Response to Reviewer bWys**
>
> We thank the reviewer for this detailed review. We will soon upload an updated version of our work.
>
> >Strengths: 1] I appreciate that you explore and validate the details of your LPALM parameterization (i.e. section 3.1), which seems to often get left out in unrolling works.
>
> Thank you for this positive comment.
>
> >Your related works section is good [...] Novel Deep Architectures"
>
> We thank the reviewer for these suggestions, which are indeed relevant in the non-blind context. We will add them in text.
>
> >in section 1.2 you say that "[NBSS] is limited by the fact that most methods assume A* to be known and fixed...". Isn't that the definition of NBSS?
>
> Yes, this is indeed the definition of NBSS, but this nevertheless constitutes a limitation in some experimental settings, such as the one we consider here in astrophysics, in which $A^{\*}$ is unknown. We will rephrase this statement in the article to make it clearer.
>
> >I don't think you broke any rules, [...] different training paradigm that confounds your comparison.
>
> We understand that the article might be a bit dense for the reader. In the future version of our article, we will simplify the experimental setup so that it is easier to understand. According to the comments of the different reviewers, we will furthermore move details towards the appendices.
> Concerning the training of our competitors' methods, it is true that the training paradigm is different. Indeed, the unrolled algorithms used for HSU are all designed to be trained and tested on a single hyperspectral image (HSI) X. This is inherently different since their goal is to unmix this single image. To do so, a part of the image pixels $\{x_i \in {\mathbb{R}^{m\times 1}},i \in [1,N]\}$ $(N<t)$ is chosen to be the training set , and the model is tested on the entire image pixels $\{x_i \in {\mathbb{R}^{m\times 1}},i \in [1,t]\}$. This makes sense since problem (1) in our paper is equivalent to $t$ problems of the form: $x_i=A^*s_i^{\*}+ n_i, i \in [1,t]$, where $x_i \in \mathbb{R}^{m \times 1}, A^{\*} \in \mathbb{R}^{m \times n}, s_i^{\*} \in \mathbb{R}^{n \times 1}$ and $n_i \in \mathbb{R}^{m \times 1}$. In contrast, LPALM takes each image $X$ as a training or testing instance. The data is thus a set of images, not pixels. In addition, to the best of our knowledge, we are the first to propose performing unrolled BSS in the semi-supervised setting (e.g. the training is done on data sets which entail $A$ matrices different from the ones in the test set). Therefore, it is difficult to find other recent unrolled algorithms with the same training paradigm.
> >In Fig 3b, instead of showing the intermediate results [...] a longer recurrent net?).
>
> You are right. In the future version of our work, we will include a plot with the results of different sized LPALMs.
>
> >The reason I gave a 2/4 in "Correctness" [...] you even mention that learning rate schedule was a "difficult" hyperparameter to set in Sec 3.3)
>
> This comment is interesting. Maybe we were unclear in the text: our learning rate choice was easy, since we are in the semi-supervised setting. For instance, we use the same constant learning rate $10^{-4}$ for all the article experiments, without requiring a learning rate schedule. On the other hand, our competitors' algorithms (in particular SNMF-net and MNN-BU) were more difficult to tune because the algorithm is unsupervised. Furthermore, they rely on a good initialization and the learning rate must be carefully chosen so that their output stays close to the initialization. Such points will be made clearer in the future version of our work.
> >To be clear, unrolling is fully understood to give improvements [...] who knows about this application.
>
> To the best of our knowledge, we are the first to unroll the PALM algorithm in a supervised way for source separation. We believe that our approach is very interesting in the field of source separation, as the LPALM algorithm enables to obtain much better results than its competitors. Concerning the training paradigm, we understand your apprehension, but we want to highlight that performing other comparisons is not straightforward, as most unrolled source separation algorithms have a different training procedure.

---

### Official Review · Reviewer_7bHu · 2021-11-03

**Correctness:** 2
**Technical Novelty And Significance:** 2
**Empirical Novelty And Significance:** 2
**Recommendation:** 5
**Confidence:** 2

**Main Review:**

The approach presented in the paper is interesting, but several important details are missing or insufficiently explained. As a result, it is hard to judge the quality of the proposed algorithm and its potential impact. First, the issues with the existing algorithms are never well explained. When it comes to the original (unlearned) PALM, the authors state that it issues include “high sensitivity of the estimated factors with respect to the choice of regularization parameters” and that “the solution depends strongly on initialization”. What do these issues look like and what is causing them? Why do we expect that introducing a learned component would improve results here? Prior to this, the authors introduce the objective function in equation (2), but do not explain it. For example, what is the role of the oblique constraint that makes up the third term in the function?

The authors go on to introduce “semi-blind” source separation. While this is an interesting idea, it is hard to understand what is meant exactly. From the later experiments, it seems that the authors rely on access to synthetic data to provide a set of potential mixing and source matrices that can be used to train the network. It is not clear, however, how the proposed method would work in the absence of such additional information. Additionally, the source matrices are often generated by simple Gaussian sampling, so it is not obvious what additional structure is retained here.

When describing the actual algorithm in Section 2, the authors go back and forth between various proposed steps for the algorithm before settling on a specific configuration. It is again hard to follow their argument it is never spelled out in detail what the resulting algorithm looks like. This is complicated by notation, like the uppercase pi in equation (6), that is introduced but never defined.

For their numerical experiments, the authors first concentrate on the source-matrix update step. They test various algorithms for this and conclude that one of them (LISTA-CP) performs better than the others and is chosen for inclusion in the proposed LPALM algorithm. However, this is a preliminary experiment and no attempt is made to follow this up by comparing the performance of the full LPALM algorithm with the various types of source matrix update steps. The authors also make statements like “LISTA trainable parameters … tend to be optimized by marginalizing over the varying mixing matrices A* in the training set, which could have led to deteriorated results”. What does this mean in the context of the proposed algorithm? Here and elsewhere in the paper, there are several steps missing in the explanations.

The authors then go on to compare LPALM with PALM and other state-of-the-art algorithms. It is found to perform much better than the rest, but it is hard to judge these results since the setup and methods are not very well explained. There are some additional details in the appendix, but overall, it is not clear what is happening. For example, why can the other methods only be tested on a single image? What does that mean in this context? For LPALM, the training data is synthetic – how does this affect results? Furthermore, when testing on real data, how is the “ground truth” in Figure 4(a–d) obtained?

There are also spelling errors (“taylored”), grammatical errors (“less” → “fewer”, “allowed to opt”, “a further interest to use”, and so on), unclear formulations, and unfortunate formatting choices (heavy use of underlining and boldface) that make the paper hard to read. I suggest the authors look over the text carefully before resubmitting. There are also many errors in the bibliography with missing journals, dates, and incorrectly capitalized titles.


**Summary Of The Paper:**

EDIT: I thank the reviewers for their explanations and updates to the paper and have updated my score accordingly.

This paper introduces a method for sparse semi-blind source separation, which aims to separate signals into a linear combination of mixing components. The components are given by an unknown mixing matrix and source (or coefficient) matrix. The goal is to estimate both the mixing and source matrices from data. Since this is an ill-posed problem, certain sparsity priors are used to regularize the problem. However, this is often not sufficient and additional constraints are needed.

To solve this, the authors introduce an unrolled version of the proximal alternating linearized minimization (PALM) algorithm, where various matrices and parameters in the algorithm are learned through backpropagation in a differentiable programming framework. Some numerical results on synthetic and experimental data are provided, illustrating the potential of the approach.


**Summary Of The Review:**

The paper proposes an interesting method with promising results, but there are several problems in the presentation of the algorithm, its motivation, and the numerical results. The main issue is a lack of clarity, which undermines the results presented. As such, I do not recommend that this paper be accepted for publication.

---

> ### Author Response · Authors · 2021-11-18
> **Response to Reviewer 7bHu [1/2]**
>
> We thank the reviewer for this detailed review. We will soon upload an updated version of our work.
> >What do these issues look like and what is causing them?
>
> Such issues have been in-depth studied in *[1] Kervazo, C., Bobin, J., Chenot, C., \& Sureau, F. (2020). Use of PALM for $\ell_1$ sparse matrix factorization: Difficulty and rationalization of a two-step approach. Digital Signal Processing, 97, 102611.*, and we therefore build directly on this work. In particular, their study highlights that PALM obtains worse separation qualities than other optimization algorithms (for instance projected alternating least squares). They explain that this is due to the high sensitivity of PALM final estimate with respect to the initialization strategy (because the problem is non-convex) and the hyper-parameter choice (because the gradient step does not fully invert the matrix $A$ at each iteration, contrary to a least square step). We refer to their work for more details.
> >Why do we expect that introducing a learned component would improve results here?
>
> This is a good question, and we will add the following clarifications within the article:
> - As [1] emphasizes the high impact of the cost function smooth part's optimization strategy (specifically, they emphasize the differences between the algorithm estimates when gradient steps or least-squares are used), we propose in this work to learn the gradient steps to try to improve the unmixing and make it faster.
> - More generally, using learning in LPALM enables to learn a new (implicit) regularization of the sought-after $A$ and $S$ factors, which is inferred from the training set.
>
> >For example, what is the role of the oblique constraint that makes up the third term in the function?
>
> This cost function is very common in blind source separation and, due to lack of space, we refer the reader to classical resources such as *Comon, P., \& Jutten, C. (Eds.). (2010). Handbook of Blind Source Separation: Independent component analysis and applications. Academic press.*, which explains in detail from where it comes. We will nevertheless add a sentence to explain the oblique constraint. In short, such a constraint enables to avoid degenerated solutions in which $\|\|S\|\|_F \rightarrow 0$ and $\|\|A\|\|_F \rightarrow +\infty$.
> >The authors go on to introduce “semi-blind” source separation. While this is an interesting idea, it is hard to understand what is meant exactly. From the later experiments, it seems that the authors rely on access to synthetic data to provide a set of potential mixing and source matrices that can be used to train the network.
>
> The reviewer is right : we assume to have access to synthetic data to provide a set of potential mixing and source matrices that can be used to train the network. This is true in many applications: the one we consider here in astrophysics, but also in remote sensing in which, for instance, spectral libraries are often available. In the future version of our work, we will include the above explanation.
> >It is not clear, however, how the proposed method would work in the absence of such additional information.
>
> We thank the reviewer for this interesting comment. We precisely want to highlight that the presence of such additional information is key to obtain good estimates, and this is a part of the novelty of our approach. Indeed, we show in the numerical section that other methods such as SNMF, which work in a fully unsupervised fashion, fail to obtain good separation qualities.
>
> >Additionally, the source matrices are often generated by simple Gaussian sampling, so it is not obvious what additional structure is retained here.
>
> Precisely, we used a Generalized Gaussian distribution, which is a broader family than the Gaussian distribution. Our numerical results tend to show that our method works well even when the training set for $S$ is composed of samples following this simple law. This is actually a very positive thing, as it means that the method does not seem to require highly sophisticated source S simulators to obtain good results. In this case, the structure that is retained is rather related to the mixing matrix $A$: jointly with the input data $X$, it enables the network to obtain good estimates for $S$.
> > When describing the actual algorithm in Section 2, the authors go back and forth between various proposed steps for the algorithm before settling on a specific configuration. It is again hard to follow their argument it is never spelled out in detail what the resulting algorithm looks like.
>
> The algorithm updates are spelled in (8), in the *LPALM section*. In the future version, we will add a box around the equations to highlight them.
>
> >This is complicated by notation, like the uppercase pi in equation (6), that is introduced but never defined.
>
> This will be corrected. It corresponds to the projection on the $\ell_2$ unit ball of each column of $A$.

---

> > ### Author Response · Authors · 2021-11-18
> > **Response to Reviewer 7bHu [2/2]**
> >
> > >For their numerical experiments [...] there are several steps missing in the explanations.:w
> >
> > We thank the reviewer for this interesting comment. While preparing this work, we actually tried the various updates in the LPALM algorithm. These experiments were not included due to lack of space, but according to your comment we will add them in the future article version.
> > We will furthermore clarify the quoted sentence in the new version of our work: as LISTA reparametrization motivation is to set $W_1=\frac{1}{L}A^{\*T}A^{\*}$ and $W_2=\frac{1}{L}A^{\*T}$, the LISTA parameters are expected to be $A^*$-dependent: they should rather be denoted as $W_1(A^*)$ and $W_2(A^*)$. If they are learnt over a training set in which $A^*$ is not constant, the learnt $W_1$ and $W_2$ (over the whole training set) should resemble to an aggregate of the different $W_1(A^*)$ and $W_2(A^*)$. This is what we mean by the sentence “LISTA trainable parameters … tend to be optimized by marginalizing over the varying mixing matrices A* in the training set". Now, this can lead to deteriorated results for a given test sample $X_{test} = A^*_{test}S^*_{test} + N_{test}$ since it is likely that the learnt $W_1 \neq W_1(A^*_{test})$ and $W_2 \neq W_2(A^*_{test})$.
> >
> > >The authors then go on to compare LPALM with PALM and other state-of-the-art algorithms. It is found to perform much better than the rest, but it is hard to judge these results since the setup and methods are not very well explained. There are some additional details in the appendix, but overall, it is not clear what is happening. For example, why can the other methods only be tested on a single image? What does that mean in this context?
> >
> >  We will explain this point more precisely in the future version of our work.
> > To summarize, the unrolled algorithms used for HSU are all designed to be trained and tested on a single hyperspectral image (HSI) X. This is inherently different since their goal is to unmix this single image. To do so, a part of the image pixels $\{x_i \in {\mathbb{R}^{m\times 1}},i \in [1,N]\}$ $(N<t)$ is chosen to be the training set , and the model is tested on the entire image pixels $\{x_i \in {\mathbb{R}^{m\times 1}},i \in [1,t]\}$. This makes sense since problem (1) in our paper is equivalent to $t$ problems of the form: $x_i=A^{\*}s_i^{\*}+ n_i, i \in [1,t]$, where $x_i \in \mathbb{R}^{m \times 1}, A^{\*} \in \mathbb{R}^{m \times n}, s_i^{\*} \in \mathbb{R}^{n \times 1}$ and $n_i \in \mathbb{R}^{m \times 1}$.
> > In contrast, LPALM takes each image $X$ as a training or testing instance. The data is thus a set of images, not pixels.
> >
> > >For LPALM, the training data is synthetic – how does this affect results?
> >
> > The experiment we do in section 3.3 (3.2 in the new version) highlights the fact that even if the sources are generated with a simple Generalized Gaussian law, the results of LPALM when applied to a data set with real sources are very good. This is to the credit of LPALM, which thus does not require to have access to highly precise simulations of the sources to obtain good results.
> >
> > >Furthermore, when testing on real data, how is the “ground truth” in Figure 4(a–d) obtained?
> >
> > The ground-truth of the sources was obtained from a real data set by the Generalized Morphological Component Analysis algorithm (GMCA), which is currently state-of-art in sparse source separation. Two remarks are in order:
> > - As we generated the data $X$ from these sources, it is important to note that these sources can really be considered as a ground truth, although they were initially estimated by GMCA.
> > - Applying our algorithm directly on a data $X$ would make the comparisons between the algorithms very difficult due to the absence of the ground truth. Therefore, we believe that using GMCA to generate the data set is relevant.
> >
> > >There are also spelling errors (“taylored”), grammatical errors (“less” → “fewer”, “allowed to opt”, “a further interest to use”, and so on), unclear formulations, and unfortunate formatting choices (heavy use of underlining and boldface) that make the paper hard to read. I suggest the authors look over the text carefully before resubmitting. There are also many errors in the bibliography with missing journals, dates, and incorrectly capitalized titles.
> >
> > This will be corrected.
> >
> > >The paper proposes an interesting method with promising results, but there are several problems in the presentation of the algorithm, its motivation, and the numerical results. The main issue is a lack of clarity, which undermines the results presented. As such, I do not recommend that this paper be accepted for publication.
> >
> > We thank the reviewer for this feedback. We hope that the above explanations have clarified our work with respect to your comments. Most of them will be included in the future version of our work.

---

> ### Author Response · Authors · 2021-12-03
> **Please let us know if our responses have addressed your concerns**
>
> Dear reviewer 7bHu,
>
> Thank you again for your valuable review, which was very constructive and helped us refine our article. Please let us know if our responses have addressed your concerns. We will be ready for any additional clarification.
>
> Best.

---

### Official Review · Reviewer_rsVe · 2021-11-04

**Correctness:** 3
**Technical Novelty And Significance:** 3
**Empirical Novelty And Significance:** 2
**Recommendation:** 6
**Confidence:** 4

**Main Review:**

The main contributions are the following:
- An extension of unrolled alternating minimization is given that preserves the structure of the variable A
- Experiments on both synthetic and real multispectral imaging data show improved results compared to established baselines

Pros:
The formulation of LPALM is interesting and practical. By explicitly representing A in the unrolls, the structure of A is directly used and can adapt at test-time. There is also significant time savings using the unrolled approach compared to the standard AltMin of PALM.

Cons:
While the extension of LISTA to LPALM is intellectually interesting, there are a number of other works that also investigate unrolled alternating minimization on a sparse S and a general A, for example with (convolutional) sparse coding. There are also works that extend this to deep learning based-regularization. See for example:
[1] Tolooshams, S. Dey, and D. Ba, “Deep residual autoencoders for expectation maximization-inspired dictionary learning,” IEEE Transactions on Neural Networks and Learning Systems, pp. 1–15, 2020.o
[2] Marius Arvinte, Sriram Vishwanath, Ahmed H. Tewfik, Jonathan I. Tamir, "Deep J-Sense: Accelerated MRI Reconstruction via Unrolled Alternating Optimization." MICCAI (6) 2021: 350-360

Notably, in these works, the supervised loss is only applied on the S variable. Therefore, it is unclear what value/tradeoff exists between assuming a specific A (which is estimated from the data and known to be inaccurate), vs. a true semi-blind loss function.

It is also not clear how the proposed method would compare to a simpler version of learned PALM, such as "LPALM-LLT", which would be similar to the ISTA-LLT while still updating A through alternating minimization. Therefore, it is not clear which component is the driving force for improved results.

There is also no theoretical justification for the choice of algorithm, and no analysis of convergence.

I also had difficulty understanding what data is actually real and non-simulated, vs. realistic but simulated data. I do not understand the comments in Appendix B related to the preprocessing. Perhaps the authors could make this more clear.

Based on this assessment, I believe the paper is marginally below the acceptance threshold.

The justifications for this decision is the limited number of experimental results and comparisons, missing ablations (such as supervised on S alone, or unsupervised as presented in Xiong et al.


**Summary Of The Paper:**

Edit -- score has been updated to reflect the revisions by the authors. Remaining review below is unchanged.

This paper studies the linear mixing inverse problem, also known as blind source separation, in the context of multispectral imaging. Specifically, the work considers the situation X = AS + N, where S is assumed sparse and A has some unknown structure. The authors consider unrolling an alternating minimization based on ISTA, and learning hyperparameters and weight matrices of the unrolled optimization, which they call LPALM. Improved results are shown compared to standard AltMin (called PALM), as well as simpler unrolled methods such as LISTA that do not explicitly account for the structure of A. The learning is done using a training set where A and S are decently known and can be used as a supervised loss.


**Summary Of The Review:**

In summary, the paper proposes an extension to LISTA, called PALM, which unrolls an alternating minimization to solve a bilinear inverse problem, i.e. sparse coding. The experiments indicate that the proposed approach is an improvement over conventional alternating minimization, but more experimental work is needed as well as proper contextualization with other unrolled methods related to unrolled blind deconvolution, dictionary learning, and other bilinear inverse problems.

---

> ### Author Response · Authors · 2021-11-18
> **Response to Reviewer rsVe [1/2]**
>
> We thank the reviewer for this detailed review. We will upload in the next few days an updated version of our work.
> >Cons: While the extension of LISTA to LPALM is intellectually interesting, there are a number of other works that also investigate unrolled alternating minimization on a sparse S and a general A, for example with (convolutional) sparse coding.
>
> Just to ensure that we made our contribution clear, please note that LPALM is slightly more than an extension of LISTA : 1) it uses an alternating structure between the two variables $A$ and $S$ (while LISTA only considers the estimation of $S$) ; 2) It uses specific update forms (such as LISTA-CP for $S$), which enables in particular the algorithm to alternate between the variables and to obtain better results.
> >There are also works that extend this to deep learning based-regularization. See for example: [1] ... [2] ...
>
> The cited work [1] indeed investigates unrolled learning of A and S, but important differences with ours have to be highlighted:
> - It focuses on convolutional sparse coding, while our work aims at performing source separation. Although these two problems share mathematical similarities, their goals are in general very different: we here aim at recovering *ground truth dictionaries* and *sparse codes*. In contrast, dictionary learning algorithms generally target specific tasks (such as denoising) in which the dictionary is not important by itself: what matters is the dictionary's ability to help achieving the task at hand (for instance, to obtain well denoised images). That being said, the cited article considers a numerical experiment in the context of source separation. Nevertheless, the experimental setting is very simple (only two sources), and the initialization seems to be very close to the algorithm results. Therefore, we rather focused on comparisons with algorithms that have truly been designed for source separation (SNMF-net, MNN-NU and DNMF), since they are likely to obtain better results.
> - From a methodological point of view, the cited work is very different from ours. Dictionary learning is tackled through an auto-encoder. The encoder part enables to estimate S, while the decoder enables to estimate A and the hyperparameter $\lambda$. As such, there is only a single alternation between A and S, in contrast to our method in which each layer enables to update both $A$ and $S$.
> - In the cited work, no unrolling is used for $A$ and $\lambda$ updates. Furthermore, the encoder structure, which enables to estimate $S$, is derived through an unrolling of FISTA. In contrast, we use a LISTA-CP update, enabling several alternations between $A$ and $S$.
>
> The application in [2], MRI, is very different. This impacts the algorithm, which is also different: for instance, in MRI the sampling is done in the Fourier space, which must be taken into account in the algorithm. The algorithm also considers convolutional problems, in contrast to ours. In the cited work, some neural networks are used to replace the proximal operators. In contrast, we propose to learn some parameters within the gradient, which is completely different.
> Therefore, since the approach is fully different, as well as the application, it is unclear how the cited algorithm should be adapted to our astrophysics source separation context.
>
> Here again, we preferred to perform comparisons with algorithms that have been tailored for source separation. Nevertheless, if the reviewer finds the above answers interesting, we can include part of them, as well as the two cited articles, in the future version of our work.
> >It is unclear what value/tradeoff exists between assuming a specific A (which is estimated from the data and known to be inaccurate), vs. a true semi-blind loss function.
>
> We indeed think that this is an important question. The goal of section 3.1 is actually to try to assess such a tradeoff. As can be seen from Figure 2, the results of non-blind algorithms (LISTA, LISTA-CP) are actually deteriorated when $A^*$ is not perfectly estimated. This is the motivation for the LPALM algorithm. Beyond these experiments, it would be difficult to directly compare LPALM and most non-blind algorithms (for instance ISTA or LISTA-CP) when $A^*$ is not perfectly known, because the results of non-blind algorithms highly depend on $A^*$ estimate quality (cf section 3.1).
> Please note that a noticeable exception to this is the case of LISTA, which does not require explicitly the $A^*$ matrix as an input. Nevertheless, LISTA performed much worse than LPALM and we therefore focused the comparisons to other algorithms designed for source separation. If the reviewer however believes that the numerical comparison of LPALM and LISTA is interesting, we can include it in the future version of our work.

---

> > ### Author Response · Authors · 2021-11-18
> > **Response to Reviewer rsVe [2/2]**
> >
> > >It is also not clear how the proposed method would compare to a simpler version of learned PALM, such as "LPALM-LLT", which would be similar to the ISTA-LLT while still updating A through alternating minimization. Therefore, it is not clear which component is the driving force for improved results.
> >
> > The reviewer is fully right. Although we did such tests during the preparation of our work, and our architecture obtained the best results, we did not include them due to lack of space. This will be corrected in the future version of our work. To summarize such experiments, LPALM performs better than its simpler counterparts.
> >
> > >There is also no theoretical justification for the choice of algorithm, and no analysis of convergence.
> >
> > This is true, although we still aimed at proposing a quite in-depth understanding of our algorithm, and in particular of the unrolling choices we did (this is specifically the purpose of Section 3.1, which aims at focusing on LPALM S-update to better understand its behavior).
> >
> > >I also had difficulty understanding what data is actually real and non-simulated, vs. realistic but simulated data. I do not understand the comments in Appendix B related to the preprocessing. Perhaps the authors could make this more clear.
> >
> > We understand that the different data sets might cause a difficult understanding. If time allows, we will try to clarify this in the future version of our work by using a single data set in all the article.
> > Concerning the pre-processing, the take home message is that we use a wavelet transform and focus on the detail scales (which are sparser). Once $A$ has been estimated from them, we estimate the sources coarse scale by applying the pseudo-inverse of $A$ to the coarse-scale of the data $X$. Such a practice is quite usual in sparse source separation, and we therefore did not originally include them in the text. This will be updated in the future version of our work.
> >
> > >The justifications for this decision is the limited number of experimental results and comparisons, missing ablations (such as supervised on S alone, or unsupervised as presented in Xiong et al.
> >
> > In the new version of our work, we will add an ablation study and several other comparisons for instance, with different update choices in LPALM.
> > Prior to that, we would like to comment on the ablation study:
> > - We want to highlight that we aimed at proposing an algorithm which does not require a difficult fine-tuning of its parameters. In particular, a more complicated version could incorporate some weights before each of the losses on $S$ and $A$ in the loss function (9). Specific cases would include a supervision on $S$ alone or $A$ alone. Nevertheless, it would conduct to design choices that might be specific to a given data set, which we wanted to avoid as much as possible. We therefore do not claim our cost function (9) to be the best possible one (maybe supervision on $S$ enough might be better) but to be quite generic. In addition, such a generic cost function already enables LPALM to largely outperform its competitors.
> > - Concerning an unsupervised cost function, a first rough answer can be given considering SNMF-net, which is an unsupervised algorithm sharing some similarities with LPALM. Our algorithm largely outperforms it, which is a first hint to answer that having an unsupervised cost function might lead to worse results. To more precisely answer the reviewer point, we will add in the future version of our work the results of LPALM with an unsupervised cost function. The figures (that we already have) show that LPALM performs much worse when trained within an unsupervised setting.

---

> > > ### Comment · Reviewer_rsVe · 2021-11-29
> > > **Updated score to reflect the changes**
> > >
> > > Thank you to the authors for incorporating significant changes to the paper to make it easier to follow and clarify different components. I have updated my score to reflect the changes.

---

### Author Response · Authors · 2021-11-29
**General comments on the new version of the article**

We would like to thank the reviewers for their valuable comments. In order to answer to some of their questions, several changes have been made in the new version of the article. We here summarize the most important ones:
- The paper was reorganized to make it clearer. In particular, an experiment was postponed to the appendices to enable us to gain space and make the article easier to read.
- We modified the \"related works" section and added the references suggested by the reviewers.
- We added a small discussion about why introducing some learning within PALM is expected to be interesting.
- The algorithm was more clearly stated in the text. We further included in Section 2.1 a paragraph on LPALM wavelet preprocessing. The whole algorithm (LPALM + preprocessing) is summarized in Appendix A.
- We added in Section 2.2 an explanation of the differences between LPALM and its competitors training methods.
- We explained more clearly whether synthetic or realistic data is used. We further reduced the number of data sets from 3 to 2, to make the experimental setting clearer. Appendix B was also modified to be more understandable.
- In order to make the experiment "_Which model is empirically the best for updating S in the presence of variabilities over A*?_"  more convincing, we increased the number of sources to $4$. The number of layers was furthermore increased to $25$, to answer to a reviewer comment. Please note that due to lack of space, this experiment is reported to Appendix D.1.
- In Appendix D.2, we added an experiment in which the different update parametrizations (LISTA-CP, ISTA-LL...) are tested _inside_ the alternating structure of the PALM algorithm.
- We rewrote section 3.1.1 (about PALM initialization issue) to make it more convincing. We further tested PALM with several different initializations, instead of a single one, to give more significance to the experiment.
- In section 3.2 (realistic experiment), we further added LISTA results. It enables us to further demonstrate the interest of alternating between the two A and S updates within LPALM.
- We showed in Appendix D.4 the impact of the number of layers on the final NMSE of LPALM.
- We made in Appendix D.5 an ablation study on the loss function used for LPALM. This further confirms the fact that merely using as a cost function a data-fidelity term (unsupervised learning) leads to bad solutions.
- We corrected grammatical and syntax errors, reformulated some sentences, and corrected the errors in the bibliography (missing journals, incorrectly capitalized titles...).

---

### Author Response · Authors · 2021-12-09
**Thank you**

Thanks to all the reviewers for having evaluated the new version of the paper and updated the global scores.

For the correctness, technical novelty and significance, empirical novelty and significance scores, we would be grateful if the reviewers update them accordingly, in the case they find that it is relevant. Thank you again for your precious time and effort!

---

### Decision · Program_Chairs · 2022-01-20

**Decision:**

Accept (Poster)

**Comment:**

The paper develops an unrolled version of the PALM algorithm for sparse blind (or semi-blind) source separation. The unrolled version includes a soft-thresholding update, in which the thresholding parameter and one of the weight matrices is learned from data, with a least squares dictionary update, in which the step size is learned from data. The paper provides experimental results showing that this LPALM algorithm is less sensitive to the choice of hyper parameters (since step sizes, etc. are learned from data), and to the choice of the initial dictionary (perhaps since the W matrices are learned from similar examples). It also improves over PALM on experimental data from an astronomy problem.

Reviewers expressed appreciation for the paper’s experimental results, and detailed investigation of the parameterization of unrolled PALM. They also highlighted some issues in the initial submission's exposition -- in particular, the setting of the problem (what kind of training data is available, what is the relationship between the mixing matrices A at training and at test time), and a clearer explanation of why it makes sense to learn fixed matrices W^{(k)} which do not depend on A (given that A may change at test time). The revision improved the clarity of the paper, addressing most of these concerns. The submission contributes to the discussion on how to unroll dictionary learning / blind source separation algorithms, how the unrolled algorithm should be parameterized, and demonstrates good results on multispectral data analysis.